psychology

video games, aggression, violence

**Author for correspondence:**
Christopher J. Ferguson
e-mail: cjferguson1111@aol.com

# Do longitudinal studies support long-term relationships between aggressive game play and youth aggressive behaviour? A meta-analytic examination

Aaron Drummond[1], James D. Sauer[2]
and Christopher J. Ferguson[3]

[1]Massey University, Palmerston North, Manawatu-Wanganui, New Zealand
[2]Department of Psychology, University of Tasmania, Hobart, Tasmania
[3]Department of Psychology, Stetson University, 421 North Woodland Boulevard, DeLand, FL 32729, USA

CJF, 0000-0003-0986-7519

Whether video games with aggressive content contribute to aggressive behaviour in youth has been a matter of contention for decades. Recent re-evaluation of experimental evidence suggests that the literature suffers from publication bias, and that experimental studies are unable to demonstrate compelling short-term effects of aggressive game content on aggression. Long-term effects may still be plausible, if less-systematic short-term effects accumulate into systematic effects over time. However, longitudinal studies vary considerably in regard to whether they indicate long-term effects or not, and few analyses have considered what methodological factors may explain this heterogeneity in outcomes. The current meta-analysis included 28 independent samples including approximately 21 000 youth. Results revealed an overall effect size for this population of studies ($r = 0.059$) with no evidence of publication bias. Effect sizes were smaller for longer longitudinal periods, calling into question theories of accumulated effects, and effect sizes were lower for better-designed studies and those with less evidence for researcher expectancy effects. In exploratory analyses, studies with more best practices were statistically indistinguishable from zero ($r = 0.012$, 95% confidence interval: −0.010, 0.034). Overall, longitudinal studies do not appear to support substantive long-term links between aggressive game content and youth aggression. Correlations

between aggressive game content and youth aggression appear better explained by methodological weaknesses and researcher expectancy effects than true effects in the real world.

## 1. Introduction

Debates regarding the impact of aggressive video games on youth aggression have raged for decades, since the earliest cabinet and Atari 2600 console games. Despite almost four decades of research, no consensus has been reached among scholars [1]. Over 100 experimental studies have been conducted on aggressive game effects, but recent scholarship suggests that publication bias led to overstated confidence in the short-term outcomes for aggression that could be reliably demonstrated [2]. If short-term effects are difficult to demonstrate, this still leaves the potential for long-term effects which may accumulate over time. Thus, an examination of longitudinal data may help elucidate whether long-term effects may develop, even if short-term effects are minimal.

To date, a little over two dozen longitudinal studies have examined relationships between aggressive game play and later aggressive behaviour among youth. Outcomes from these studies have been heterogeneous. Some have found evidence for small but statistically significant relationships between game play and later aggression [3,4]. Others have concluded that no such effects exist [5,6]. Still others are more nuanced, suggesting that tiny relationships may exist, but lack clinical utility [7]. Overall, meta-analytic results suggest that effect sizes across studies are very low, typically less than $r = 0.10$ [8,9]. How such small effects should be interpreted also remains controversial, with some scholars proposing small effects are important in isolation or cumulatively over time [9], while others argue that small effects should not be interpreted as hypothesis supportive, given the high potential for misleading results owing to methodological noise [10]. One issue that has not been examined is how cross-study heterogeneity may be explained by methodological issues that influence effect sizes (e.g. use of standardized, validated aggression measures, use of proper control variables, using more than one respondent, etc.). For instance, effect sizes may be reduced towards zero once important third variables are controlled [11] or effect sizes may become inflated owing to participant hypothesis guessing, or researcher degrees of freedom allowing scholars to reanalyse their results multiple ways to find best fits with their hypotheses, much as appeared to have happened with experimental studies [12].

One recent meta-analysis [9] specifically examined longitudinal studies. Their analysis revealed a tiny but statistically significant relationship between aggressive game play and youth aggression ($r = 0.078$ with controlled effect sizes). The authors concluded these effects indicate meaningful long-term relationships. However, this analysis is limited by several factors. First, their analysis made no effort to distinguish whether several well-known limitations of video game research might have spuriously influenced the observed effect sizes. These influences include unstandardized measures of aggression, lack of preregistration, potential demand characteristics and non-independence of violent content ratings for aggressive games (e.g. asking participants to rate the violent content of the games they play). Without careful methodological control, weak effect sizes such as $r = 0.078$ might reflect methodological issues rather than the effects of interest. Second, several longitudinal studies appear to have been missed in the analyses [6,13]. Further, effect sizes from several studies reported in the meta-analysis may not have been an accurate representation of the effect sizes from those studies. In fairness, it is possible the authors may have asked individual study authors for specific analyses difficult for us to replicate. However, in some studies, it appeared that effect sizes reported by Prescott *et al.* [9], were higher than those reported in the original studies [14,15]. In other cases, a request by us for effect sizes by authors of the original studies resulted in effect sizes that differed from those reported by Prescott *et al.* [5,16]. In one case, original data were lost and reported effect sizes could not be confirmed or replicated [17]. Third, the authors of the meta-analysis appear to rely on statistical significance (which is almost always achieved in meta-analysis owing to the large number of observations), rather than a careful examination of effect sizes. Lastly, several further longitudinal studies [7,18], including two studies with preregistered analyses [19,20], have been published since Prescott *et al.*'s meta-analysis was undertaken.

Therefore, there is value in undertaking another specific examination of longitudinal studies of aggressive game influences. The current analysis sought to improve upon previous work in several ways. First, the current meta-analysis was preregistered. As with preregistration of individual studies, preregistration of meta-analyses can help mitigate questionable researcher practices (conscious or unconscious) that can undermine the integrity of the analyses. As with individual studies, meta-analytic authors may make decisions that, unwittingly, favour a particular outcome.

Preregistration can reduce this potential. Pre-registering our meta-analysis, and specifying our methodological and analytical choices prior to commencing, attenuates the risk that reported findings reflect post-hoc decisions about the treatment of the literature and the effects sizes observed. Second, the impact of methodological issues such as unstandardized measures, clinical validity and independence of video game ratings will be considered as moderator analyses. This may help us to understand the degree to which concerns about methodological issues that have been expressed in the field have practical impact on effect size outcomes. Third, the current meta-analysis includes more recent longitudinal studies including several with preregistered analyses. This will help us to update the current status of this research area.

Fourth, the analyses will undertake a more critical examination of effect sizes. Conventional statistical guidelines assert that an effect of $r = 0.10$ is considered small, while effects smaller than $r = 0.10$ should be interpreted as tiny or trivial effect sizes [21]. Moreover, some scholars assert that the smaller the effect sizes in a field, the less likely those findings are to reflect genuine effects of interest [22]. We therefore preregistered that we would employ a cut-off of $r = 0.10$ to differentiate results that are more likely to be practically meaningful from those that are less likely to be practically meaningful [23]. We acknowledge there is substantial scholarly debate about precisely where this line of practical meaningfulness lies (indeed in review, we had one reviewer argue for as small as $r \sim 0.04$ and another argue for as large as $r \sim 0.2$). We do not claim for $r = 0.10$ as an absolute, final cut-off for meaningful effects in this or all contexts. We acknowledge the need for future work to understand the implications of a range of effect sizes. Here we consider results below this value non-supportive of a meaningful relationship given that (i) they are, owing to their trivial size, in our opinion, unlikely to be indicative of sizeable or meaningful behavioural changes in aggression, and (ii) that methodological factors might plausibly produce tiny effects driven by non-veridical factors such as study design, methodological weaknesses or social desirability biases [10,19]. Again, we recognize that this standard may be controversial and hope that it will stimulate further dialogue on standards for effect size interpretation. Our concern is that, given the observation that video game studies have documented issues with (i) methodological flexibility and lack of standardization [12], (ii) questionable researcher practices in the operationalization of violent game play in survey studies [10], and (iii) demand characteristics [10] and an absence of checks for unreliable or mischievous responding [10], widespread methodological issues in the field may be creating a steady stream of misleading results. As put by Przybylski & Weinstein [10, p. 2] in speaking on prior meta-analyses in this realm, 'With this in mind, there is reason to think that outstanding methodological challenges might be inflating this metanalytic estimate'.

This issue is, in fact, well documented elsewhere. One recent analysis of meta-analyses found that, particularly when reliant on non-preregistered studies, meta-analyses tend to return mean effect sizes that are upwardly biased, as compared to preregistered studies [24]. Though an examination at this level has not been done for video game research yet, we observe that the majority of perhaps 10–12 preregistered studies of video game violence thus far have returned null results, e.g. [10,25,26] (a single preregistered study thus far has returned mixed results [27]), suggesting that meta-analyses which include wide numbers of non-preregistered studies are probably experiencing upward bias. The problem overlaps with the questionable researcher practice issues known to cause misleading results in other areas. Indeed, related areas of research such as social priming, themselves supported by prior meta-analyses, are now widely regarded as having been upwardly biased owing to methodological issues and questionable researcher practices [28]. Even for issues such as unreliable measures which would theoretically attenuate effect sizes, our observation is that these measures lack of standardization will more often result in the opposite, namely the potential for questionable researcher practices that inflate effect sizes [10].

Lastly, given concerns that methodological issues and researcher expectancy biases might influence effect sizes spuriously, we test for these issues explicitly in our moderator analyses, which have not been done in previous meta-analyses.

We specifically examined the following hypotheses:

(i) aggressive game play will be related to youth aggression with effect sizes in excess of $r = 0.10$;
(ii) studies with a greater number of best practices, such as those which employ standardized, clinically validated measures, independent ratings of video game content and carefully control for theoretically relevant third variables [29], will show greater effect sizes than studies with fewer best practices; and
(iii) Effect sizes will increase with greater time lag in longitudinal studies, demonstrating the accumulation of effects over time.

# 2. Method

## 2.1. Preregistration

As many analytic decisions are made during a meta-analysis, and as these analytic decisions can alter the results of an analysis, we preregistered our analysis plan prior to conducting this meta-analysis. The preregistration is available here: https://osf.io/gbx4w.

## 2.2. Inclusion criteria

We modelled our inclusion criteria on Prescott *et al.* [9]. Specifically, to be included, studies had to be longitudinal/prospective in nature with a minimum three-week time frame between the first and last waves of data collection. Studies also needed to include a measure of violent game content (not just time spent gaming in general) as a predictor variable, and aggressive or violent behaviour (not thoughts or feelings) as an outcome variable. Only behavioural outcomes were considered.

## 2.3. Selection of studies

Most studies were selected directly from those included in Prescott *et al.* [9]. Other meta-analyses (e.g. [8]) were also sourced for additional longitudinal studies missing from Prescott *et al.* [9]. Finally, we undertook a search on PsycINFO and Medline using the terms '(video game*) OR (computer game*) OR (digital game*)' AND 'longitudinal OR prospective' AND 'agress* OR viol*' as subject searches, with the exception of longitudinal/prospective which was maintained as all text. This search yielded 51 hits. After removing duplicates or those which did not meet the inclusion criteria expressed above, this resulted in the inclusion of a total of 28 independent samples. In a few cases, authors were unable to provide data that would have allowed for the calculation of an effect size. A PRISMA diagram is included as appendix A. The list of studies is available at: https://osf.io/xcvhg/.

Some datasets have produced multiple articles (e.g. the Singapore dataset [19], also [7,30]). Given such articles may use different analytical methods, effect sizes can vary considerably between them. Between studies, preference will be given for effect sizes with the maximum number of theoretically relevant controls (defined below) as well as for preregistered studies should they exist (as was the case for the Singapore dataset). Only one study from each dataset was included in the final analysis. In the event of competing publications, we included those that used standardized assessments, included the most theoretically relevant controls (i.e. gender, time 1 aggression, family environment, mental health, peer environment), or were preregistered. In the event the dataset was available, this was used to calculate effect sizes (rather than relying on published effect sizes).

## 2.4. Analysis plan

Similar to Prescott *et al.* [9], the main effect size was standardized regression coefficients (betas) which were calculated from the effect size employing the greatest degree of theoretically relevant controls in each study. Theoretically relevant variables have been identified for media violence research generally, including gender, family environment, mental health, personality and, for longitudinal studies, time 1 outcome variables [31]. Two authors extracted effect sizes from each article. We calculated interrater reliability for this to be $\alpha = 0.97$.

Initial results were calculated using comprehensive meta-analysis (CMA). CMA was used to calculate random effects weighted mean effect sizes, as well as conduct moderator analyses. Publication bias was assessed using the shinyapps meta-analysis calculator. Publication bias was assessed with tools including basic funnel plot analysis, trim and fill, precision-effect test/precision-effect estimate with standard errors (PET/PEESE), *p*-curve and *r*-index. *P*-curve analysis was preregistered to only be used if more than 20% of *p*-values were marginal, (that is, between 0.01 and 0.05). Our purpose for potentially incorporating a *p*-curve analysis was to correct for an overabundance of marginal *p*-values if one existed, which may indicate *p*-hacking or other questionable researcher practices. We therefore assumed that we should normally expect a number of marginal *p*-values by chance and thus did not intend to run a *p*-curve analysis if there did not appear to be an overabundance of marginal *p*-values. We recognize this is not the only reason for undertaking a *p*-curve analysis, but it was our intent in using it, hence our preregistration.

Given the high power of meta-analysis, almost all meta-analyses return 'statistically significant' effects. Consistent with recommendations of Orben & Przybylski [23], we considered an effect size of $r = 0.10$ the minimum for practical significance. A copy of all data is available at https://osf.io/e8pvm/.

## 2.5. Best practices analysis

We coded studies for employing current best practices (e.g. preregistration) to determine whether using such practices had an impact on the effect sizes reported. Studies were given a point each for the inclusion of a number of different best practices (see below), resulting in a numeric score that ranged from 0 to 6. This score was used as a moderator variable to determine the effect of employing best practices on effect size. Studies were given credit (1 point each) for the following best practices:

1. using a standardized outcome measure for aggression (those for which aggression is measured identically across studies without opportunity for researchers to change the measurement approach);
2. not relying on respondents to rate the violence in video games but rather using independent ratings such as scholars or ratings boards (e.g. Entertainment Software Ratings Board, Pan European Game Information). Such measures have been found to be valid indices of violent content when compared to the ratings of trained, blinded raters [32] and correlate highly with other approaches and demonstrate good construct validity [33];
3. using a clinically validated measure of aggression (e.g. Child Behaviour Checklist) such that it has been validated in a clinical setting against a clinical outcome (e.g. conduct disorder, arrests, school referrals for aggression, etc.);
4. using more than one respondent (e.g. parent and child);
5. controlled, at minimum, gender and T1 aggression; and
6. preregistration of analysis plan. Given a high degree of misleading results in non-preregistered studies [34], this appeared to be a prudent inclusion.

## 2.6. Control analysis

Papers were coded for the use of theoretically relevant controls in their analyses. This allowed an assessment of how the inclusion of theoretically relevant controls influenced effect sizes. Specifically, papers were given 1 point each for including the following theoretically relevant control variables:

1. controlling for gender;
2. controlling for T1 aggression;
3. controlling for family environment;
4. controlling for mental health; and
5. controlling for peer influences.

## 2.7. Citation bias

Papers were assessed for citation bias. To determine if a paper suffered from citation bias, we examined the literature review. If the literature review included no citations to papers with conclusions that conflict with the authors' hypotheses, they were coded as having citation bias. Papers that acknowledged at least one research study or paper conflicting with the authors' hypotheses, were coded as not having citation bias.

## 2.8. Moderator analyses

The following preregistered variables were included in moderator analyses to determine whether they influenced reported effect sizes: age of the sample, year of the study, best practices, control analysis and citation bias. For continuous moderator analyses, meta-regression was used.

# 3. Results

The overall effect size across all longitudinal studies was $r = 0.059$ (95% confidence interval (CI), 0.034, 0.084). Although 'statistically significant' ($p < 0.001$), this value is well below the preregistered threshold of $r = 0.10$. Further, between-study heterogeneity was high ($Q = 74.914$, $p < 0.001$, $I^2 = 63.96$) suggesting that this value should not be interpreted as indicative of a population effect size.

Two studies conflated video game violence with other forms of media violence [35,36]. Efforts to obtain an effect size for games only were not successful. We made the decision to include these

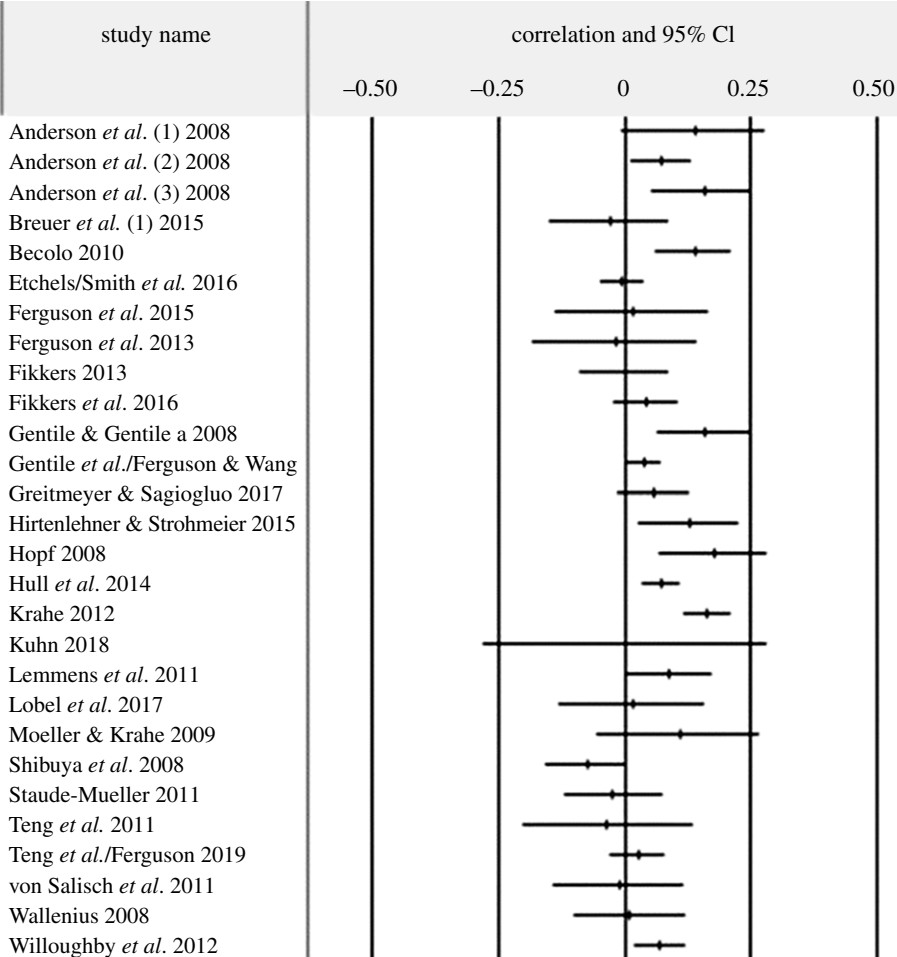

**Figure 1.** Forest plot.

studies as they did include video game violence information. However, an argument could also be made for excluding them. Excluding both studies would actually reduce the overall effect size to $r = 0.055$ (95% CI, 0.031, 0.078).

A forest plot of included studies with effect sizes is presented as figure 1.

## 3.1. Moderator analyses

Regarding moderator analyses, continuous moderators were tested using meta-regression. There was a slight trend toward studies with longer longitudinal periods having lower effect sizes than studies with shorter longitudinal periods ($Q = 5.07$, $p = 0.024$, $r = -0.046$). This conflicts with the view that effects accumulate over time. Given the small effect size, it is best interpreted as longitudinal period having little impact on study effect size. Best practices were also associated with lower effect sizes ($Q = 31.04$, $p < 0.001$, $r = -0.617$). Specific inclusion of theoretically relevant controls was not a predictor of effect sizes, however ($Q = 1.64$, $p = 0.200$).

There was a slight relationship between study year and effect sizes, such that effect sizes have got smaller in more recent studies ($Q = 9.37$, $p = 0.002$, $r = -0.33$). Participant age was not a moderator of effect size ($Q = 1.01$, $p = 0.316$).

Regarding categorical moderators, studies that suffered from citation bias reported significantly higher effect sizes ($r = 0.114$, 95% CI: 0.078, 0.150) than those without citation bias ($r = 0.032$, 95% CI: 0.007, 0.057). This difference was statistically significant ($Q = 13.51$, $p < 0.001$).

## 3.2. Publication bias

None of the publication bias analyses indicated publication bias for the full group of included studies. P-curve and the r-index did not indicate a cluster of results around $p = 0.05$, nor an unusually high proportion of positive results given observed power. Although the result for the PET–PEESE intercept

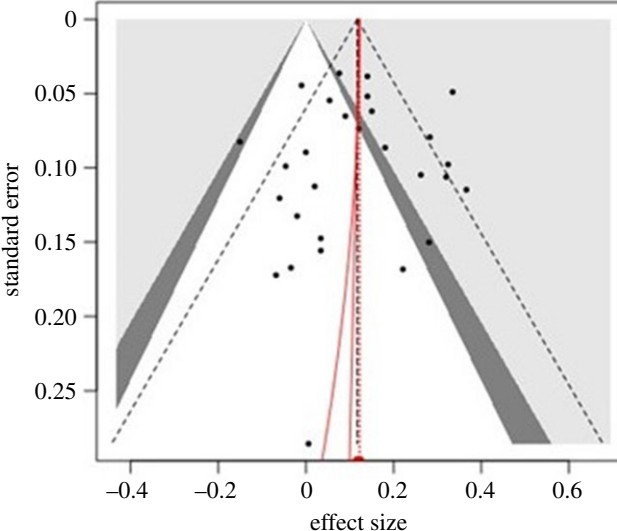

**Figure 2.** Funnel plot of included studies. Note: effect sizes here are represented in effect size *d*.

**Table 1.** Publication bias results.

| publication bias test | outcome |
| --- | --- |
| Orwin fail safe *N* | number of studies required to reduce effect size to $r = 0.10$ is 0 |
| Begg and Mazumdar rank correlation | $p = 0.492$ (one tailed) |
| Egger's regression | $p = 0.427$ (one tailed) |
| trim and fill | no missing studies |
| PET/PEESE | b0 (intercept) = 0.12, $t_{26} = 4.111$; $p < 0.001$. bias correction remained $r = 0.060$ |
| *p*-curve (*p*-hack) | $Z = 4.856$; $p = 1.000$ |
| *r*-index | **success rate** = 0.3929 |
| | **median observed power** = 0.3271 |
| | **inflation rate** = 0.0658 |
| | ***r*-index** = 0.2613 |

was statistically significant, the adjusted effect size was effectively identical to the original effect size ($r = 0.060$ versus $r = 0.059$). A funnel plot of studies is provided as figure 2. All publication bias outcomes are reported in table 1.

## 3.3. Exploratory analysis

We conducted a further analysis to examine whether ethnic groups differed in regard to effect size. These differences were non-significant ($Q = 4.029$, $p = 0.133$). Random effects analysis suggested that effect sizes were a bit higher for Whites ($r = 0.073$) than Latinos ($r = 0.015$) and Asians ($r = 0.027$), but not to a significant degree. It is also worth noting that studies of Latinos were also more likely to employ standardized validated aggression instruments such as the Child Behaviour Checklist. As such, to some degree, ethnicity is conflated with study quality.

We also examined which specific best practices had the most impact on effect sizes. Effect sizes were lower for studies which use standardized outcome measures ($Q = 16.501$, $p < 0.001$), validated outcome measures ($Q = 10.739$, $p < 0.001$) as well as independent ratings of video game content ($Q = 4.323$, $p = 0.038$). Other best practices were not associated with effect sizes. This may have been owing to low variance issues, however. Relatively few studies used multiple responders ($Q = 0.010$, $p = 0.919$), most studies did include controls for gender and T1 aggression ($Q = 0.036$, $p = 0.849$), but only two studies were preregistered ($Q = 1.691$, $p = 0.193$) (both of which were null studies.) These results are presented in table 2.

**Table 2.** Effect sizes from exploratory analyses.

| grouping variable | k | n | effect size | 95% CI | $I^2$ | tau |
|---|---|---|---|---|---|---|
| ethnicity | | | | | | |
| Whites | 20 | 13 514 | 0.073 | 0.041, 0.104 | 65.78 | 0.055 |
| Latinos | 3 | 750 | 0.015 | −0.057, 0.087 | 0.00 | 0.000 |
| Asians | 7 | 6380 | 0.027 | −0.015, 0.069 | 48.84 | 0.036 |
| standardized outcome | | | | | | |
| yes | 17 | 12 938 | 0.026 | 0.003, 0.050 | 31.57 | 0.026 |
| no | 11 | 8107 | 0.113 | 0.078, 0.147 | 48.81 | 0.038 |
| validated outcome | | | | | | |
| yes | 5 | 2920 | −0.005 | −0.042, 0.031 | 0.00 | 0.000 |
| no | 23 | 18 125 | 0.070 | 0.043, 0.097 | 63.51 | 0.048 |
| independent game ratings | | | | | | |
| yes | 12 | 8211 | 0.028 | −0.005, 0.062 | 33.49 | 0.030 |
| no | 16 | 12 834 | 0.079 | 0.045, 0.114 | 71.85 | 0.057 |

Lastly, we conducted an analysis for studies that were above the median for best practices (scoring 3 or above, $k = 10$). For this subset of studies, the mean effect size was $r = 0.012$ (95% CI: −0.010, 0.034). As the CI crosses zero, this suggests that the effect in these studies is indistinguishable from zero.

# 4. Discussion

Experimental investigations of the short-term effects of aggressive game content on player aggression produce inconsistent results [2]. As can now be seen, both an initial meta-analysis without much consideration of methodological moderators [9] and the current, updated meta-analysis suggest that effects fall below the $r = 0.10$ benchmark for a small effect. Publication bias indicators yielded no evidence of publication bias. Thus, current research is unable to support the hypothesis that violent video games have a meaningful long-term predictive impact on youth aggression. However, a number of findings merit more explicit consideration.

## 4.1. How to interpret weak effects

First, as noted, the overall effect of aggressive game content on behavioural aggression was below our preregistered cut-off for a practically meaningful effect (and the traditional cut-off to be considered small). This brings us to acknowledge one weakness of meta-analysis in general, namely the focus on statistical significance. We observe that, for years, scholars have acknowledged that 'statistical significance' is a poor benchmark for theory support [37] yet psychologists often naively rely upon it when making decisions. We argue that, particularly in highly powerful analyses such as meta-analysis, the concept of statistical significance becomes irrelevant as almost everything is statistically significant. Small effects, even where statistically significant, are often explained through methodological noise such as common method variance, demand characteristics or single-responder bias. Indeed, in our study we find that effect sizes are largely inflated through issues such as poorly standardized and validated measures of both aggression and violent game content. As such, relying on 'statistical significance' can give scholars an inflated sense of confidence in their hypotheses and render the concept of 'effect size' little more than window dressing, where any effect size, no matter how small, can be interpreted as supporting the hypothesis.

We acknowledge that our adoption of the $r = 0.10$ standard is likely to stimulate debate, which we believe to be important and welcome. Although we adopted the 0.10 standard suggested by Przybylski & Weinstein [10], one of the authors has previously suggested that an even higher standard of 0.20 may be necessary for greater confidence in the validity of effects [38] though the origins of such concerns about over-reliance on statistical significance and over-interpretation of weak effects stretches back decades. As expressed by Lykken [39, p. 153] 'the effects of common method are often as strong as or stronger than those produced by the actual variables of interest'. This raises the question of to what degree we can have confidence that observed effect sizes reflect the relationship of interest as opposed to research artefacts. To be fair, some

scholars do argue for interpretation of much lower effect sizes, such as $r = 0.05$ [40], though it is important that a key phrase in this argumentation is noted: 'Our analysis is based on a presumption that the effect size in question is, in fact, reliably estimated' [40, p. 163]. Our observation is that this assumption appears to have been demonstrated to be false for this field of research and, with that in mind, a higher threshold of scrutiny is warranted. Funder and Ozer's argument also relies on effects accumulating over time, whereas our analysis found the opposite, that longer time-intervals were associated with smaller effect sizes. Our concerns are less about the issue that some effects may be of trivial importance (though there is that), but rather that some observed effect sizes do not index genuine effects of interest at all, being instead the product of systematic methodological limitations. Naturally, we do not suggest that our $r = 0.10$ threshold is the end of this debate. Further data may suggest that this number needs be revised either upwards or downwards (we suspect the former more likely than the latter). Standards may need to be flexible given the differences in rigor across different fields, or even across prior assumptions about the size of effect one expects to see. For example, there is a rough precedent for $r = 0.10$ from another meta-analysis of aggression and empathy wherein weak effect sizes results ($r = 0.11$) were interpreted as not hypothesis supportive [41]. The authors note this could be owing to either weaknesses in the theory or measurement problems (or both) and we agree those are both worthwhile issues to consider. Our observation that standardized and validated measures tend to produce weaker effects for this field, however, would appear to diminish the possibility for measurement attenuation as a driving factor of observed weak effect sizes. This cautious interpretation of weak effect sizes has precedent, as well, among meta-analyses of violent games. For instance, an earlier meta-analysis found larger effect sizes ($r = 0.15$), but based on methodological and theoretical issues identified in the field, interpreted this as non-convincing [42].

The adoption of the $r = 0.10$ standard also appears consistent with the 'smallest effect sizes of interest' (SESOI) approach. From this perspective, an SESOI can be developed based on multiple criteria including what is theoretically relevant, what prior literature has suggested is an important effect size, what effect sizes are considered important based on established (though ultimately arbitrary) criteria, and the degree to which resources may be burned on an effect without seeing tangible outcomes [43]. From this approach, we can see the $r = 0.10$ standard is defensible. Both Orben and Przybylski, as well as earlier standards set by Cohen [21], apply the 0.10 standard (though we acknowledge other scholars endorse either higher or lower standards). Further, previous meta-analyses have suggested effects should be in the range of 0.20–0.30 [44], so any observed effects under 0.10 would represent an appreciable decline in effect size. Lastly, as we observe significant methodological issues have the potential to inflate effect size estimates, as also noted by Przybylski and Weinstein, setting an interpretive threshold can help reduce misinterpretation of weak, possibly misleading results. We note our CI does not cross the 0.10 threshold and, as such, feel confident in interpreting that the threshold for interpreting the longitudinal data as meaningful has not been met.

These debates regarding the interpretation of small effect sizes exist in other realms as well. This is particularly true when large samples may result in many 'statistically significant' relationships between variables that bear little actual relationship to each other. For instance, one recent study linked emotional diversity to mental and physical health in two samples totalling 37 000 relying on effect sizes generally in the range of $r = 0.10$ and lower (some as low as $r = 0.02$) [45]. Reflecting our concerns here, this interpretation was criticized by other scholars who argued such weak findings were more like the product of statistical artefacts than genuine effects of interest [46]. Regarding the potential perils of misleading results in large samples, the first author of that critique states (N. Brown 2020, personal communication) 'A large sample size is a good thing, but only if used to improve the precision of effect estimates, not to prove that cats equal dogs, $p < 0.05$'. We agree with this assessment. Naturally, our critique is not of the use of large samples, which we wholeheartedly endorse, but rather the lack of consideration for potential statistical 'noise' (demand characteristics, single-responder bias, common method variance, researcher expectancy effects, mischievous responding, etc.), and how these can cause misleading results (for a specific, discovered example, see [47]).

Alternatively, the issue could be considered from the 'crud factor' perspective of Meehl [48]. From this perspective, tiny effects are real though only in the sense that every low-level variable is correlated to every other low-level variable to some degree (i.e. the $r = 0.00$ is rarely strictly true). This alternative explanation returns the dialogue to that of triviality. If every variable is correlated to every other variable to a tiny degree and in a way that will become statistically significant in large samples, it is still valuable to understand which relationships rise above this 'crud' and are worthy of investigation or policy interventions. Otherwise, the argument that video games might be restricted to promote youth mental health may be no more critical than, quite literally, arguing for the restriction of potatoes or eyeglasses for the same reason [49].

We welcome debate on this issue and challenges to our own position. We believe that this is a discussion worth having and one which extends beyond video game research.

## 4.2. Other issues

Second, we find no evidence for the assertion that these small effects might accumulate over time. The negative association between the length of the longitudinal period and the size of the effect speaks directly against theories of accumulated effects. Indeed, we found that longer longitudinal periods were associated with smaller effect sizes, not larger, directly contradicting the accumulation narrative. This is consistent with older meta-analyses of experimental studies which, likewise, found that longer exposure times were associated with weaker effects [42]. However, this differed from a previous meta-analysis that found some evidence for a positive association between longitudinal time and effect size [9]. However, the Prescott et al. meta-analysis found this effect only for fixed effects analyses, not for random effects, and random effects would probably have been the more appropriate model given heterogeneity in the data. Further, longitudinal time was treated as a 3-part categorical variable rather than a more appropriate continuous variable, and this may have caused statistical artefacts. Our analysis also includes several newer longitudinal studies not included with Prescott et al. As to why longitudinal length is associated with reduced effect size, we can think of two categories of explanation. First, there is a genuine, small effect of interest, but this is relatively short-lived and does not accumulate. Second, there is not a genuine effect of interest and methodological issues such as demand characteristics or mischievous responding tend to have greater impact on short-term outcomes than long. Given our observations about widespread methodological limitations in this field and their impact on effect size, we suspect the latter option is more likely. As such, for this area at least, we recommend against using this narrative, and suspect it should be used more cautiously in other areas as well unless directly demonstrated through empirical studies.

Third, we demonstrate that study quality issues do matter. In particular, the use of standardized and well-validated measures matters. Specifically, the use of high-quality measures was associated with reduced effect sizes. This observation also undercuts claims in a previous meta-analysis for ethnic differences in video game effects [9]. In particular, studies with Latino participants were mostly done with a population from Laredo, Texas, and used highly validated measures such as the Child Behaviour Checklist. Thus, Latino ethnicity was conflated with highly validated, standardized aggression measures. Only one other study involved Hispanics, Caucasian Americans and African Americans, but this study both used a non-standard video game assessment, and an unstandardized aggression measure, and switched the psychometrics of the aggression measure in the final time point, making longitudinal control difficult [50]. As such, this study is not of a quality sufficient to examine for ethnic differences. Given there are no obvious reasons to think Latinos are *immune* to game effects, whereas non-Latinos are *vulnerable*, it is more parsimonious to conclude that differences in the quality of measures used were responsible for the observed ethnic differences.

Studies which were of higher quality (scoring above the median in best practices) returned an effect size statistically indistinguishable from zero. This suggests that some effects may be driven by lower quality practices that may inflate some effect sizes. It is worth noting too that issues such as the use of standardized and validated outcomes and other best practices tend to correspond with less citation bias. As such, concerns about best practices and researcher allegiances tend to overlap.

Lastly, studies evidencing citation bias had higher effect sizes than those that did not demonstrate citation bias. This may be an indication of researcher expectancy effects. As such, we recommend increased use of preregistration in empirical studies.

It is worth noting that our effect size of $r = 0.059$ may not appear very different from the controlled effect size of $r = 0.078$ obtained by Prescott and colleagues. However, this represents a reduction in explained variance from 0.608% to 0.348%. This is a reduction of approximately 43% of explained variance (0.348/0.608). Granted this reduction would seem more dramatic had the original figure of $r = 0.078$ been larger. As meta-analyses on violent video games have been repeated over time, there appears to be a consistent downwards tendency in their point estimates, declining from 0.15 in early estimates to 0.10 in uncontrolled and 0.078 in controlled longitudinal estimates. Our results show further reduction towards an effect size of 0 in a preregistered longitudinal meta-analysis employing theoretically relevant controls. Despite disagreement about where the precise line of the smallest effect of interest may be, downward trends in the meta-analytic point estimates over time suggest we need to, as a field, grapple with precisely where an effect becomes too small to be considered practically meaningful or risk overstating the importance of our findings.

Further, we observe that few studies were preregistered. Preregistration can be one means by which research expectancy effects can be reduced. Consistent with observations about upward bias in meta-analyses in other realms, it appears that, across study types, preregistered analyses have been much less likely to find results in support of the video game violence hypotheses than non-preregistered studies. Our meta-analysis is, to our knowledge, the first preregistered meta-analysis in this realm. Preregistration is seldom perfect, of course, and we recognize there will always be some debate and subjectivity in terms of extracting the ideal effect size that best represents a hypothesis; but preregistration of both individual studies and meta-analyses can help make decisions clearer and reduce researcher subjectivity at least partially.

At this juncture, we observe that meta-analytic studies now routinely find that the long-term impacts of violent games on youth aggression are near zero, with larger effects sizes typically associated with methodological quality issues. In some cases, overreliance on statistical significance in meta-analysis may have masked this poor showing for longitudinal studies. We call on both individual scholars as well as professional guilds such as the American Psychological Association to be more forthcoming about the extremely small observed relationship in longitudinal studies between violent games and youth aggression.

Data accessibility. All data are available at: https://osf.io/e8pvm/.

Authors' contributions. A.D., J.D.S. and C.J.F. all contributed equally to the conceptualization of the paper, selection of included studies, data analysis and writing of the final manuscript.

Competing interests. We declare we have no competing interests.

Funding. Supported by the Marsden Fund Council from Government funding, managed by Royal Society Te Apārangi; MAU1804.

# Appendix A

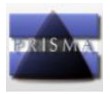 PRISMA 2009 flow diagram

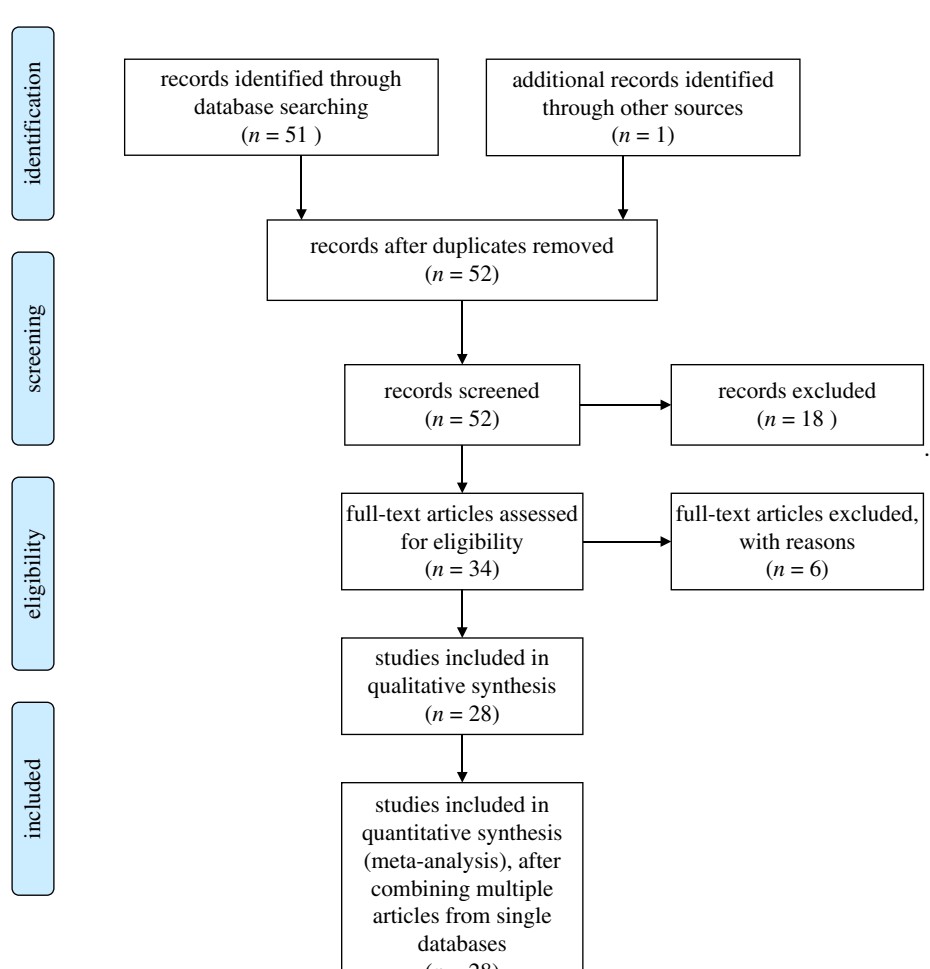

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

**13**