## [Reviewer comments · Royal Society Open Science]

Review History

RSOS-192165.R0 (Original submission)

Review form: Reviewer 1

Is the manuscript scientifically sound in its present form?

No

Are the interpretations and conclusions justified by the results?

No

Is the language acceptable?

Yes

Do you have any ethical concerns with this paper?

No

Have you any concerns about statistical analyses in this paper?

Yes

Recommendation?

Reject

Comments to the Author(s)

The authors present a meta-analysis of longitudinal studies of violent video game play and subsequent aggression. They conclude: "Overall, longitudinal studies do not support long-term links between aggressive game content and youth aggression." In doing so, they level several criticisms at a previously published meta-analysis by Prescott et al. (2018).

Despite the authors' criticism of the meta-analysis reported by Prescott et al., the present results are very similar to those reported by them. Thus, Prescott et al. report a random effects model mean effect $r = .078$, 95% CI = .053,.102. The current authors report a mean effect $r = .06$, 95% CI = .035, .084. Note that the lower bound of the Prescott et al. estimate covers the mean effect reported by the current authors and the upper bound of their estimate covers the Prescott et al. mean effect.

As written, I have multiple problems with the data on which the authors conduct their meta-analysis and a variety of their specific claims.

1. There are multiple large and small errors in the data used in this meta-analysis. Referring to author citations in the supplemental table:

a. Breuer et al (1). In order to obtain this estimate, the authors apparently combined two independent studies (one with $N = 140$, one with $N = 136$) into one data point (with $N = 276$). This is inappropriate and renders the meta-analytic mean effect estimate and confidence intervals inaccurate.

b. Etchells/Smith et al. The Etchells et al paper is on conduct disorder (rather than physical aggression). It reports results for 1804 participants. The results in the Etchells et al. Table 2 are reported in terms of odds ratios. The fully adjusted result for the six level measure of conduct disorder is positive and statistically significant ($p = .05$). The same is true for the two level case status measure ($p = .044$). I have no idea where the current authors obtained a non-significant negative effect, how they obtained an N of 2019, or where the Smith et al. analysis enters the picture.

c. The estimates for Ferguson (2012) and Ferguson et al. (2013) differ from those reported by Prescott et al. (although the respective N s are the same). Prescott et al. used the Youth Self Report (YSR) effects in Table 2 for Ferguson (2012) and similarly the YSR effects in Table 1 for Ferguson et al (2013). I could not find the exact results reported in the current meta-analysis in either original paper.

d. The Prescott et al. meta-analysis uses Youth Self Report data from Ferguson et al. (2011). The latter paper is excluded from the present analysis, presumably because the data overlap those used in the Ferguson et al. (2012) and/or Ferguson et al. (2013) studies. On the other hand, neither the 2012 nor the 2013 paper acknowledge that their data overlap those used in the 2011 paper, leading the reader to assume they are independent subsamples and observed effects. Indeed, both papers cite Ferguson (2011) as an independent source validating the general approach (Ferguson et al. 2012, p. 142; Ferguson et al. 2013, p. 112) and in correspondence with Prescott et al. the corresponding author represented these findings as independent. If these are not independent studies, it should have been made clear in each publication and all correspondence.

e. Fikkers et al. (2013). This is a study of media violence. Results for violent electronic game play is not reported separately. Until that is done, including these data in the meta-analytic estimate of the effects of violent video games is inappropriate.

f. Fikkers et al. (2016). The Prescott et al. estimate was obtained from data provided by Karin Fikkers (personal communication, April 17, 2017). The current meta-analytic estimate is different and it is not clear where the authors got it.

- g. Gentile & Gentile (2008a, 2008b). I don't know where the current authors got these specific numbers.
- h. Greitmeyer & Sagiogluo (2014). The results reported in Prescott et al. were obtained directly from Tobias Greitemeyer (personal communication April 24, 2017). The current authors' estimate is different and it is not clear where they got it.
- i. Hirtenlehner & Strohmeier (2015). The effect size estimate reported in Prescott et al. was obtained directly from Helmut Hirtenlehner. The current authors' estimate is different and it is not clear where they got it.
- j. Hull et al. (2014). The current authors' N is incorrect. The N as stated in Prescott et al is 2723. Apparently, the authors simply summed the White, Hispanic, and Asian subsample Ns in the Hull et al (2014) study instead of using the overall N (which included Black participants) or contacting the authors. The effect size should be .075 as reported in Prescott et al. (in other estimates, the current authors use three decimal accuracy, but not here).
- k. Krahe et al. (2012). The results reported in Prescott et al. were obtained directly from Krahe's co-author Robert Busching (personal communication October 30, 2014). The current authors' estimate is different and it is not clear where they got it unless they just averaged values in Table 3.
- l. Lemmens et al. (2011). The N used by the current authors is incorrect. Lemmens et al. (2011) started with an initial sample of 851 as stated in the abstract, but as both the Method and the Results sections make clear, the results are based on 540 participants.
- m. Lobel et al. (2017). The N used by the current authors is incorrect and does not take into account missing data that affected the analysis. As reported in Table 1 of Lobel et al., the N on which the longitudinal analyses are based should be 184.
- n. Möller & Krahe (2009). The Prescott et al. estimate is based on the direct and indirect effects of video game play in the structural model (while covarying any effect associated with gender). The current authors apparently base their estimate only on the direct effects, inappropriately dropping the mediated effects – presumably considered to be noise that should be covaried out of the effects of game play. Indirect effects are actually more informative than direct effects because they elucidate the intervening psychological processes whereby violent video games have their effects. As such, correct estimates for the effect of any exogenous variable should be based on both direct and indirect effects. The corresponding author has been criticized in published work for failing to recognize this basic methodological fact.
- o. Shibuya et al. (2008). The Prescott et al. parameter estimate and sample N are based on a personal communication from Shibuya (July 11, 2014). I have no idea where the current authors obtained their parameter estimate. The authors' N is apparently from the reported total sample, but does not take into account missing data encountered in the analyses.
- p. Staude-Müller (2011). The N used by Prescott et al. was obtained from the Staude-Müller analyses. The current authors' N is apparently from the reported total sample, but does not take into account missing data encountered in the analyses.
- q. von Salisch et al. (2011). The Prescott et al. N was obtained from the von Salisch analyses (see their Figure 2). The current authors' N is apparently from the reported total sample, but does not take into account missing data encountered in the analyses.

As can be seen, the number of errors in the current authors' meta-analytic data set, both in terms of parameter estimates and sample Ns, is sizeable. They are also partially responsible for the differences between their analysis and that of Prescott et al. If the above errors are corrected, the random effects model mean $r = .070$ (95% CI .046, .095) – actually closer to Prescott et al than the model mean estimate presented in this study.

2. Prescott et al. are criticized for excluding two studies used by these authors. A study by Teng et al. (2011) did not meet the Prescott et al. selection criteria of measuring the level of violent video game exposure and physical aggression at an initial point in time. A study by Wallenius and Punamäki (2008) was excluded because the authors inappropriately included a Time 2 covariate (parent child communication) in their analysis of the effect of Time 1 Game Violence on Time 2 aggression. The latter covariate was the single strongest predictor of time 2 aggression

(other than time 1 aggression), but has the unfair advantage of being measured coincident with the dependent variable.

The current authors excluded a study by Adachi and Willoughby (2016) because the “original data were lost” (p. 4). Prescott et al. received these data directly from Teena Willoughby (personal communication, September 26, 2016).

If the two studies excluded by Prescott et al. are dropped from the corrected data in #1 above, the random effects mean model $r = .074$ (95% CI .049, .099), even closer to the .078 finding reported by Prescott et al. Adding the study by Adachi and Willoughby (2016) simply narrows the confidence interval slightly, $r = .074$ (95% CI .050, .098).

3. The above (#1 and #2) reduced analyses still include the authors’ new estimates for (a) Etchels/Smith et al. (2016), (b) Gentile et al (2014)/Ferguson & Wang (2019), (c) Kuhn (2018), (d) Lobel et al. (2017), and (e) Teng et al./Ferguson (2019).

It is important to note, however, that the new data added by the authors include two sizeable samples of Asian participants. Ferguson & Wang (2019) report results for 3034 participants from Singapore (and replaces the Gentile et al. (2014) sample of 2029 from the same participant pool used by Prescott et al.). Ferguson (2019) reports results for 1340 participants from China. The authors fail to mention, however, that a major point made by the Prescott et al. meta-analysis is the potential moderating effect of ethnicity. Specifically, the latter authors noted that Asian samples yielded lower effect estimates than did White samples. Ironically (and unacknowledged), the Ferguson & Wang (2019) effect estimate of .038 and the Ferguson (2019) effect estimate of .027 is consistent with this argument (and brings down the overall mean effect size in a manner consistent with the ethnicity arguments of Prescott et al.). Thus, it is not surprising that the current meta-analysis obtains a mean effect size estimate lower than that observed by Prescott et al. when over 40% of the new observations are specifically selected from populations the latter authors specified as showing lessened effects.

4. Finally, despite finding a statistically significant effect of roughly similar magnitude to that reported by Prescott et al., the current authors essentially try to define it out of existence: “Overall, longitudinal studies do not support long-term links between aggressive game content and youth aggression” (abstract) and “Results below this value [of $r = .10$] will be considered non-supportive of a meaningful relationship...” (p. 5). The authors then specifically hypothesize “Aggressive game play will be related to youth aggression with effect sizes in excess of $r = .10$ ” (p. 5) in order to subsequently reject it. It is not clear why the authors even pose this as a hypothesis in light of the findings reported by Prescott et al.

5. The authors state: “One issue that has not been examined is how cross-study heterogeneity may be explained by methodological issues that influence effect sizes” (p. 3). This is false. Prescott et al. specifically examined the moderating effect of selected methodological issues (participant ethnicity, study time lag, participant age).

6. The authors state: “For instance, effect sizes may be reduced toward zero once important third variables are controlled...” (p. 3). Prescott et al. specifically examined estimates that included versus did not include covariates beyond the initial time lag and found the difference to be minimal. This study confirms this observation (see p. 10).

7. The authors specifically criticized the meta-analysis by Prescott et al. in several regards.

a. “...several longitudinal studies appear to have been missed in the analyses” (p. 4). The authors cite two: Teng et al. (2011) and Wallenius and Punamäki (2008). As noted above, Prescott et al. did not miss these studies, they excluded these studies on specific methodological grounds (see 2 above).

b. “...in some studies it appeared that effect sizes reported by Prescott et al., were far higher than those reported in the original studies” (p. 4). The authors provide one citation: Ferguson et al. (2012). Prescott et al. used an effect size of .03 for this study. This value appears in Ferguson et al. (2012) Table 1 for the variable of interest.

c. “In other cases, a request by us for effect sizes by authors of the original studies resulted in effect sizes far lower than those reported by Prescott et al” (p. 4). The authors provide one citation: Fikkers et al. (2016). As noted above (1f), the Prescott et al. effect size was provided directly from Karin Fikkers in a personal communication (April 17, 2017).

d. All of these criticisms and the above responses beg the question: Why didn't the authors directly write to the corresponding author of Prescott et al. and ask for clarification of these issues instead of directly, and in my opinion unjustifiably, attacking their work? Indeed, in conducting their research Prescott et al. had written directly to the corresponding author of the current work, incorporated data that he had shared, and cited work conducted by him in their interpretation of observed effects.

8. With respect to the data selected for the analyses, the authors state: “Only one study from each dataset was included in the final analysis” (p. 7). Unfortunately, it is not at all clear which studies were excluded and why.

9. It has always struck me the “best practices” analyses such as those reported by the authors (as well as “control analyses” and “citation bias” analyses) are subject to the biases of the authors. At the very least, the authors should point out that “best practices” analyses conducted by other authors have found exactly the opposite effects (and, ironically, have been criticized by the current corresponding author as biased).

10. The authors report “a tendency for the literature to under-represent null findings” (p. 12). Such claims of selection bias have been repeatedly debunked – including by the current study (p. 11).

11. The authors do not report full citations for the studies included in the meta-analysis. Indeed, they do not even report full author lists. This makes it extremely difficult for interested readers to duplicate or critically evaluate their work.

Review form: Reviewer 2

Is the manuscript scientifically sound in its present form?

Yes

Are the interpretations and conclusions justified by the results?

Yes

Is the language acceptable?

Yes

Do you have any ethical concerns with this paper?

No

Have you any concerns about statistical analyses in this paper?

No

Recommendation?

Accept with minor revision (please list in comments)

Comments to the Author(s)

This paper is well-written, interesting, and solid in the methods and conclusions. I recommend acceptance. I do have a few minor comments.

1. The flow of the statement of problem on p. 5 is choppy. I suggest reorganization.
2. I agree with using the effect size of $r = .10$ (Orben & Przybylski, 2019) for this paper, but I seriously doubt if an effect size lower than $r = .20$ actually has any practical significance. This is an observation, not a suggestion. (p. 8)
3. I know that preregistration of analysis plan is very popular, but I'm not certain that it is necessary to credit this in the "best practices" calculation (p. 8).
4. Controlling for gender and T1 aggression is credited both in the best practices analysis (p. 8) and the control analyses (p. 9). This seems redundant.

Review form: Reviewer 3 (Netta Weinstein)

Is the manuscript scientifically sound in its present form?

Yes

Are the interpretations and conclusions justified by the results?

Yes

Is the language acceptable?

Yes

Do you have any ethical concerns with this paper?

No

Have you any concerns about statistical analyses in this paper?

No

Recommendation?

Major revision is needed (please make suggestions in comments)

Comments to the Author(s)

This paper presents a meta-analysis of longitudinal studies relating violent gaming to aggression. Although much attention has been given to experimental effects on aggression, relatively little has been done to understand the accumulation of the literature on its long-term effects. This is a problem for two reasons. First, experimental studies tell us little, one way or another, about long-term consequences of violent games. Second, discussion among parents and policy-makers primarily revolves on the latter, not the former. The authors make a strong argument for the need of this meta-analysis. Admittedly, I'm not an expert at meta-analytic methodology, but I also was struck by the rigor with which this research was conducted, particularly the very useful coding system for determining previous study quality.

So the following comments are aimed at improving the framing of the manuscript, because I think the practical significance of the result, $r = .06$, does not come through as clearly as it could.

I would like to know a bit more about the importance of $r = .10$ as a cutoff. The authors propose that it is a benchmark, even - a qualitatively different one, for interpreting meaningful versus meaningless results. This is a serious claim! Could the authors speak to the need and practical meaning of this effect size? For example, what other conclusions have been made at $r = .11$ that could not be made at $r = .08$.

Furthermore, the current meta-analysis did not find an effect much different than the one identified by Prescott et al. (.078) despite a number of more rigorous methodological choices made in the current study. In both cases, the effect fails the proposed .10 benchmark, right? Given this, it might be helpful to invest more time helping readers to understand the trouble with

interpreting .078 effect size as being meaningful, as in the Prescott meta-analysis, and treating this more accurately as a replication using more conservative methodology and novel moderator effects. What did the previous authors conclude that wasn't substantiated by data; why should we be concerned that they made these conclusions? The authors do these things to some extent already, but it is buried in other criticisms of the previous meta-analysis.

Could the authors clarify - what does it mean for a meta-analysis to be pre-registered? Why should we care about this? It's easier to understand its importance of pre-registration in an experimental study, but a bit of reader education about open meta-analysis procedures would be great here.

This is my ignorance about meta-analysis, but shouldn't interrater reliability for extracting effect sizes be 1.00? In other words, what would lead to any disagreement about this relatively fact-based process?

The coding of study best practices, control analysis, and citation bias is great. It's probably one of the most important contributions of this study to the broader psychological literature outside of gaming effects. I wonder if the authors could test for the influence of specific research practices (rather than lumping them together)? I realize that the sample sizes may be too small for some of these, but even without statistical tests, providing a table with the effect sizes in studies that received a point for achieving X versus studies that did not receive such a point, would be really fascinating.

Is the moderation for Prescott et al. a function of the studies the authors of the previous meta-analysis selected to include or a particular of effects they extracted (because of bias?), or is it because additional studies have been conducted recently? That is, would it still be attained even controlling for study recency? Please consider either leaving this analysis out, or more clearly (and immediately) contextualizing this finding for readers.

The publication bias section is very brief. I wonder if the authors could describe the analyses and/or statistics involved, or, if they do not have space, do so in a supplemental analysis that could be referred to?

In the Discussion section, the focus is once again on the Prescott meta-analysis. From reading the Introduction, I am still left with the feeling that the current meta-analytic study is the first to carefully meta-analyze the current literature on longitudinal violent game effects. From reading the Method and Results, I took away that a study was conducted well, and yet once again (like Prescott) found an effect size below the proposed threshold of .10 (although I was never really convinced I should care about .10 - see comment above). I felt the lengthy reminder that this study is better than Prescott's in the Discussion section only detracted from the novelty and importance of the current study. I would be more inspired by a discussion taking into account that two meta-analytic studies (one preliminary and one carefully conducted) found statistically significant but meaningless long-term effects, and the implications of those effects for understanding violent games.

I hope these comments are useful, and look forward to seeing a next draft of the manuscript.

Netta Weinstein

Decision letter (RSOS-192165.R0)

10-Feb-2020

Dear Dr Ferguson,

Manuscript ID RSOS-192165 entitled "Do Longitudinal Studies Support Long-Term Relationships Between Aggressive Game Play and Youth Aggressive Behavior? A Meta-Analytic Examination" which you submitted to Royal Society Open Science, has been reviewed. The comments from reviewers are included at the bottom of this letter.

In view of the criticisms of the reviewers, the manuscript has been rejected in its current form. However, a new manuscript may be submitted which takes into consideration these comments.

Please note that resubmitting your manuscript does not guarantee eventual acceptance, and that your resubmission will be subject to peer review before a decision is made.

Your resubmitted manuscript should be submitted by 09-Aug-2020. If you are unable to submit by this date please contact the Editorial Office.

on behalf of Dr Christina Demski (Associate Editor) and Essi Viding (Subject Editor)
openscience@royalsociety.org

Associate Editor Comments to Author (Dr Christina Demski):

The manuscript was reviewed by three reviewers and two of these highlight issues that need addressing before this manuscript can be considered for publication. The first reviewer also notes some issues with the data that was selected for the meta-analysis and this needs to be addressed before publication can be considered. The third reviewer suggests refocusing the paper on what it can add to the literature rather than framing it mostly around a critique of a previous meta-analysis which has similar findings to the current analysis.

Reviewers' Comments to Author:

Reviewer: 1
Comments to the Author(s)

The authors present a meta-analysis of longitudinal studies of violent video game play and

subsequent aggression. They conclude: “Overall, longitudinal studies do not support long-term links between aggressive game content and youth aggression.” In doing so, they level several criticisms at a previously published meta-analysis by Prescott et al. (2018).

Despite the authors’ criticism of the meta-analysis reported by Prescott et al., the present results are very similar to those reported by them. Thus, Prescott et al. report a random effects model mean effect $r = .078$, 95% CI = .053,.102. The current authors report a mean effect $r = .06$, 95% CI = .035, .084. Note that the lower bound of the Prescott et al. estimate covers the mean effect reported by the current authors and the upper bound of their estimate covers the Prescott et al. mean effect.

As written, I have multiple problems with the data on which the authors conduct their meta-analysis and a variety of their specific claims.

1. There are multiple large and small errors in the data used in this meta-analysis. Referring to author citations in the supplemental table:
 - a. Breuer et al (1). In order to obtain this estimate, the authors apparently combined two independent studies (one with $N = 140$, one with $N = 136$) into one data point (with $N = 276$). This is inappropriate and renders the meta-analytic mean effect estimate and confidence intervals inaccurate.
 - b. Etchells/Smith et al. The Etchells et al paper is on conduct disorder (rather than physical aggression). It reports results for 1804 participants. The results in the Etchells et al. Table 2 are reported in terms of odds ratios. The fully adjusted result for the six level measure of conduct disorder is positive and statistically significant ($p = .05$). The same is true for the two level case status measure ($p = .044$). I have no idea where the current authors obtained a non-significant negative effect, how they obtained an N of 2019, or where the Smith et al. analysis enters the picture.
 - c. The estimates for Ferguson (2012) and Ferguson et al. (2013) differ from those reported by Prescott et al. (although the respective N s are the same). Prescott et al. used the Youth Self Report (YSR) effects in Table 2 for Ferguson (2012) and similarly the YSR effects in Table 1 for Ferguson et al (2013). I could not find the exact results reported in the current meta-analysis in either original paper.
 - d. The Prescott et al. meta-analysis uses Youth Self Report data from Ferguson et al. (2011). The latter paper is excluded from the present analysis, presumably because the data overlap those used in the Ferguson et al. (2012) and/or Ferguson et al. (2013) studies. On the other hand, neither the 2012 nor the 2013 paper acknowledge that their data overlap those used in the 2011 paper, leading the reader to assume they are independent subsamples and observed effects. Indeed, both papers cite Ferguson (2011) as an independent source validating the general approach (Ferguson et al. 2012, p. 142; Ferguson et al. 2013, p. 112) and in correspondence with Prescott et al. the corresponding author represented these findings as independent. If these are not independent studies, it should have been made clear in each publication and all correspondence.
 - e. Fikkers et al. (2013). This is a study of media violence. Results for violent electronic game play is not reported separately. Until that is done, including these data in the meta-analytic estimate of the effects of violent video games is inappropriate.
 - f. Fikkers et al. (2016). The Prescott et al. estimate was obtained from data provided by Karin Fikkers (personal communication, April 17, 2017). The current meta-analytic estimate is different and it is not clear where the authors got it.
 - g. Gentile & Gentile (2008a, 2008b). I don’t know where the current authors got these specific numbers.
 - h. Greitmeyer & Sagiogluo (2014). The results reported in Prescott et al. were obtained directly from Tobias Greitmeyer (personal communication April 24, 2017). The current authors’ estimate is different and it is not clear where they got it.

i. Hirtenlehner & Strohmeier (2015). The effect size estimate reported in Prescott et al. was obtained directly from Helmut Hirtenlehner. The current authors' estimate is different and it is not clear where they got it.

j. Hull et al. (2014). The current authors' N is incorrect. The N as stated in Prescott et al. is 2723. Apparently, the authors simply summed the White, Hispanic, and Asian subsample Ns in the Hull et al (2014) study instead of using the overall N (which included Black participants) or contacting the authors. The effect size should be .075 as reported in Prescott et al. (in other estimates, the current authors use three decimal accuracy, but not here).

k. Krahe et al. (2012). The results reported in Prescott et al. were obtained directly from Krahe's co-author Robert Busching (personal communication October 30, 2014). The current authors' estimate is different and it is not clear where they got it unless they just averaged values in Table 3.

l. Lemmens et al. (2011). The N used by the current authors is incorrect. Lemmens et al. (2011) started with an initial sample of 851 as stated in the abstract, but as both the Method and the Results sections make clear, the results are based on 540 participants.

m. Lobel et al. (2017). The N used by the current authors is incorrect and does not take into account missing data that affected the analysis. As reported in Table 1 of Lobel et al., the N on which the longitudinal analyses are based should be 184.

n. Möller & Krahe (2009). The Prescott et al. estimate is based on the direct and indirect effects of video game play in the structural model (while covarying any effect associated with gender). The current authors apparently base their estimate only on the direct effects, inappropriately dropping the mediated effects – presumably considered to be noise that should be covaried out of the effects of game play. Indirect effects are actually more informative than direct effects because they elucidate the intervening psychological processes whereby violent video games have their effects. As such, correct estimates for the effect of any exogenous variable should be based on both direct and indirect effects. The corresponding author has been criticized in published work for failing to recognize this basic methodological fact.

o. Shibuya et al. (2008). The Prescott et al. parameter estimate and sample N are based on a personal communication from Shibuya (July 11, 2014). I have no idea where the current authors obtained their parameter estimate. The authors' N is apparently from the reported total sample, but does not take into account missing data encountered in the analyses.

p. Staude-Müller (2011). The N used by Prescott et al. was obtained from the Staude-Müller analyses. The current authors' N is apparently from the reported total sample, but does not take into account missing data encountered in the analyses.

q. von Salisch et al. (2011). The Prescott et al. N was obtained from the von Salisch analyses (see their Figure 2). The current authors' N is apparently from the reported total sample, but does not take into account missing data encountered in the analyses.

As can be seen, the number of errors in the current authors' meta-analytic data set, both in terms of parameter estimates and sample Ns, is sizeable. They are also partially responsible for the differences between their analysis and that of Prescott et al. If the above errors are corrected, the random effects model mean $r = .070$ (95% CI .046, .095) – actually closer to Prescott et al than the model mean estimate presented in this study.

2. Prescott et al. are criticized for excluding two studies used by these authors. A study by Teng et al. (2011) did not meet the Prescott et al. selection criteria of measuring the level of violent video game exposure and physical aggression at an initial point in time. A study by Wallenius and Punamäki (2008) was excluded because the authors inappropriately included a Time 2 covariate (parent child communication) in their analysis of the effect of Time 1 Game Violence on Time 2 aggression. The latter covariate was the single strongest predictor of time 2 aggression (other than time 1 aggression), but has the unfair advantage of being measured coincident with the dependent variable.

The current authors excluded a study by Adachi and Willoughby (2016) because the "original data were lost" (p. 4). Prescott et al. received these data directly from Teena Willoughby (personal communication, September 26, 2016).

If the two studies excluded by Prescott et al. are dropped from the corrected data in #1 above, the random effects mean model $r = .074$ (95% CI .049, .099), even closer to the .078 finding reported by Prescott et al. Adding the study by Adachi and Willoughby (2016) simply narrows the confidence interval slightly, $r = .074$ (95% CI .050, .098).

3. The above (#1 and #2) reduced analyses still include the authors' new estimates for (a) Etchels/Smith et al. (2016), (b) Gentile et al (2014)/Ferguson & Wang (2019), (c) Kuhn (2018), (d) Lobel et al. (2017), and (e) Teng et al./Ferguson (2019).

It is important to note, however, that the new data added by the authors include two sizeable samples of Asian participants. Ferguson & Wang (2019) report results for 3034 participants from Singapore (and replaces the Gentile et al. (2014) sample of 2029 from the same participant pool used by Prescott et al.). Ferguson (2019) reports results for 1340 participants from China. The authors fail to mention, however, that a major point made by the Prescott et al. meta-analysis is the potential moderating effect of ethnicity. Specifically, the latter authors noted that Asian samples yielded lower effect estimates than did White samples. Ironically (and unacknowledged), the Ferguson & Wang (2019) effect estimate of .038 and the Ferguson (2019) effect estimate of .027 is consistent with this argument (and brings down the overall mean effect size in a manner consistent with the ethnicity arguments of Prescott et al.). Thus, it is not surprising that the current meta-analysis obtains a mean effect size estimate lower than that observed by Prescott et al. when over 40% of the new observations are specifically selected from populations the latter authors specified as showing lessened effects.

4. Finally, despite finding a statistically significant effect of roughly similar magnitude to that reported by Prescott et al., the current authors essentially try to define it out of existence: "Overall, longitudinal studies do not support long-term links between aggressive game content and youth aggression" (abstract) and "Results below this value [of $r = .10$] will be considered non-supportive of a meaningful relationship..." (p. 5). The authors then specifically hypothesize "Aggressive game play will be related to youth aggression with effect sizes in excess of $r = .10$ " (p. 5) in order to subsequently reject it. It is not clear why the authors even pose this as a hypothesis in light of the findings reported by Prescott et al.

5. The authors state: "One issue that has not been examined is how cross-study heterogeneity may be explained by methodological issues that influence effect sizes" (p. 3). This is false. Prescott et al. specifically examined the moderating effect of selected methodological issues (participant ethnicity, study time lag, participant age).

6. The authors state: "For instance, effect sizes may be reduced toward zero once important third variables are controlled..." (p. 3). Prescott et al. specifically examined estimates that included versus did not include covariates beyond the initial time lag and found the difference to be minimal. This study confirms this observation (see p. 10).

7. The authors specifically criticized the meta-analysis by Prescott et al. in several regards.

a. "...several longitudinal studies appear to have been missed in the analyses" (p. 4). The authors cite two: Teng et al. (2011) and Wallenius and Punamäki (2008). As noted above, Prescott et al. did not miss these studies, they excluded these studies on specific methodological grounds (see 2 above).

b. "...in some studies it appeared that effect sizes reported by Prescott et al., were far higher than those reported in the original studies" (p. 4). The authors provide one citation: Ferguson et al. (2012). Prescott et al. used an effect size of .03 for this study. This value appears in Ferguson et al. (2012) Table 1 for the variable of interest.

c. "In other cases, a request by us for effect sizes by authors of the original studies resulted in effect sizes far lower than those reported by Prescott et al" (p. 4). The authors provide one citation: Fikkers et al. (2016). As noted above (1f), the Prescott et al. effect size was provided directly from Karin Fikkers in a personal communication (April 17, 2017).

d. All of these criticisms and the above responses beg the question: Why didn't the authors directly write to the corresponding author of Prescott et al. and ask for clarification of

these issues instead of directly, and in my opinion unjustifiably, attacking their work? Indeed, in conducting their research Prescott et al. had written directly to the corresponding author of the current work, incorporated data that he had shared, and cited work conducted by him in their interpretation of observed effects.

8. With respect to the data selected for the analyses, the authors state: “Only one study from each dataset was included in the final analysis” (p. 7). Unfortunately, it is not at all clear which studies were excluded and why.

9. It has always struck me the “best practices” analyses such as those reported by the authors (as well as “control analyses” and “citation bias” analyses) are subject to the biases of the authors. At the very least, the authors should point out that “best practices” analyses conducted by other authors have found exactly the opposite effects (and, ironically, have been criticized by the current corresponding author as biased).

10. The authors report “a tendency for the literature to under-represent null findings” (p. 12). Such claims of selection bias have been repeatedly debunked – including by the current study (p. 11).

11. The authors do not report full citations for the studies included in the meta-analysis. Indeed, they do not even report full author lists. This makes it extremely difficult for interested readers to duplicate or critically evaluate their work.

Reviewer: 2

Comments to the Author(s)

This paper is well-written, interesting, and solid in the methods and conclusions. I recommend acceptance. I do have a few minor comments.

1. The flow of the statement of problem on p. 5 is choppy. I suggest reorganization.
2. I agree with using the effect size of $r = .10$ (Orben & Przybylski, 2019) for this paper, but I seriously doubt if an effect size lower than $r = .20$ actually has any practical significance. This is an observation, not a suggestion. (p. 8)
3. I know that preregistration of analysis plan is very popular, but I’m not certain that it is necessary to credit this in the “best practices” calculation (p. 8).
4. Controlling for gender and T1 aggression is credited both in the best practices analysis (p. 8) and the control analyses (p. 9). This seems redundant.

Reviewer: 3

Comments to the Author(s)

This paper presents a meta-analysis of longitudinal studies relating violent gaming to aggression. Although much attention has been given to experimental effects on aggression, relatively little has been done to understand the accumulation of the literature on its long-term effects. This is a problem for two reasons. First, experimental studies tell us little, one way or another, about long-term consequences of violent games. Second, discussion among parents and policy-makers primarily revolves on the latter, not the former. The authors make a strong argument for the need of this meta-analysis. Admittedly, I’m not an expert at meta-analytic methodology, but I also was struck by the rigor with which this research was conducted, particularly the very useful coding system for determining previous study quality.

So the following comments are aimed at improving the framing of the manuscript, because I think the practical significance of the result, $r = .06$, does not come through as clearly as it could.

I would like to know a bit more about the importance of $r = .10$ as a cutoff. The authors propose that it is a benchmark, even - a qualitatively different one, for interpreting meaningful versus meaningless results. This is a serious claim! Could the authors speak to the need and practical meaning of this effect size? For example, what other conclusions have been made at $r = .11$ that could not be made at $r = .08$.

Furthermore, the current meta-analysis did not find an effect much different than the one identified by Prescott et al. (.078) despite a number of more rigorous methodological choices made in the current study. In both cases, the effect fails the proposed .10 benchmark, right? Given this, it might be helpful to invest more time helping readers to understand the trouble with interpreting .078 effect size as being meaningful, as in the Prescott meta-analysis, and treating this more accurately as a replication using more conservative methodology and novel moderator effects. What did the previous authors conclude that wasn't substantiated by data; why should we be concerned that they made these conclusions? The authors do these things to some extent already, but it is buried in other criticisms of the previous meta-analysis.

Could the authors clarify - what does it mean for a meta-analysis to be pre-registered? Why should we care about this? It's easier to understand its importance of pre-registration in an experimental study, but a bit of reader education about open meta-analysis procedures would be great here.

This is my ignorance about meta-analysis, but shouldn't interrater reliability for extracting effect sizes be 1.00? In other words, what would lead to any disagreement about this relatively fact-based process?

The coding of study best practices, control analysis, and citation bias is great. It's probably one of the most important contributions of this study to the broader psychological literature outside of gaming effects. I wonder if the authors could test for the influence of specific research practices (rather than lumping them together)? I realize that the sample sizes may be too small for some of these, but even without statistical tests, providing a table with the effect sizes in studies that received a point for achieving X versus studies that did not receive such a point, would be really fascinating.

Is the moderation for Prescott et al. a function of the studies the authors of the previous meta-analysis selected to include or a particularly of effects they extracted (because of bias?), or is it because additional studies have been conducted recently? That is, would it still be attained even controlling for study recency? Please consider either leaving this analysis out, or more clearly (and immediately) contextualizing this finding for readers.

The publication bias section is very brief. I wonder if the authors could describe the analyses and/or statistics involved, or, if they do not have space, do so in a supplemental analysis that could be referred to?

In the Discussion section, the focus is once again on the Prescott meta-analysis. From reading the Introduction, I am still left with the feeling that the current meta-analytic study is the first to carefully meta-analyze the current literature on longitudinal violent game effects. From reading the Method and Results, I took away that a study was conducted well, and yet once again (like Prescott) found an effect size below the proposed threshold of .10 (although I was never really convinced I should care about .10 - see comment above). I felt the lengthy reminder that this study is better than Prescott's in the Discussion section only detracted from the novelty and importance of the current study. I would be more inspired by a discussion taking into account that two meta-analytic studies (one preliminary and one carefully conducted) found statistically significant but meaningless long-term effects, and the implications of those effects for understanding violent games.

I hope these comments are useful, and look forward to seeing a next draft of the manuscript.

Netta Weinstein

Author's Response to Decision Letter for (RSOS-192165.R0)

See Appendix A.

RSOS-200373.R0

Review form: Reviewer 1

Is the manuscript scientifically sound in its present form?

No

Are the interpretations and conclusions justified by the results?

No

Is the language acceptable?

No

Do you have any ethical concerns with this paper?

No

Have you any concerns about statistical analyses in this paper?

Yes

Recommendation?

Reject

Comments to the Author(s)

Please see enclosed PDF file (Appendix B).

Review form: Reviewer 4 (Joseph Hilgard)

Is the manuscript scientifically sound in its present form?

Yes

Are the interpretations and conclusions justified by the results?

No

Is the language acceptable?

No

Do you have any ethical concerns with this paper?

No

Have you any concerns about statistical analyses in this paper?

No

Recommendation?

Major revision is needed (please make suggestions in comments)

Comments to the Author(s)

The authors present a reproduction and extension of a previous meta-analysis of longitudinal cross-lags from T1 violent video game use to T2 aggressive behavior. I see value in this project, but there are some things the authors could do to make the text more accurate and more agreeable to the reader.

The main claims, as I understand them, are:

- small correlations are not meaningful or are likely to be type I errors
- some effect sizes extracted by Prescott et al. are larger than the original papers would indicate
- studies using better research practices observe smaller effect sizes

#Statistical arguments

I would prefer that you not attempt to argue for a cutoff of .10. I feel that it is begging the question a little bit. I recognize that you cite a source for this, but I doubt that it is broadly appropriate to ignore all effects smaller than $r = .10$. (I am here in disagreement with your previous reviewer 2, who seems to advocate an even higher threshold for some reason.) I would much prefer that the effect size be presented and interpreted on its own terms. I do not think it necessary or helpful to argue that .06 is approximately zero.

I was also very confused at the argument that "Preliminary evidence suggests that the false positive rate for effect sizes below $r = .10$ is very high." I did not find the provided citations helpful on this point. In what domains can you know the true effect size is zero and thus $r = .10$ is mere false positive? Alternatively, if the true effect size is $r = .10$, then that is a true effect. It is again begging the question to assume that a significant effect below .10 (or even just over .10? page 6 line 22) is a false positive rather than a true effect of equal or larger size. It is also not clear what sort of "noise" you mean in this passage -- sampling error? Meehl's "crud factor"? I think it would be best to cut this line of argument -- the paper is not strengthened by it, and it is not necessary in order to perform your analysis.

Additionally, I must object to the idea of there being such a thing as an "overpowered study" (Page 6, line 6). Render unto statistical significance the things that are statistical significance's, and unto effect size the things that are effect size's.

#Documentation of effect sizes

Some of the contention among the authors, Prescott et al., and their previous reviewer 1 seems to stem from discussions of which effect sizes are correctly scraped or most appropriate. It seems to me one useful way to address and resolve such disagreements would be to 1) note the direct quote or source within each article from which the effect size was scraped (e.g., "Table 2, Model 2, 'T1 VVG'", or "page 13, left column, paragraph 2, $r(183) = .017$ "). Perhaps it would be helpful to see the argument for why an extracted effect size is the most appropriate one. This would be particularly helpful in the case that the effect size extracted differs between Prescott et al. and the current article. Some of this already appears to be listed in the posted response letter, but it would be helpful to make this available to readers and reviewers, especially in the dataset.

I was confused by the passage on page 5, line 45, "However, in some studies...". My best understanding from the passage & provided citations is that you are suggesting that Prescott et al. 1) entered too large an effect size for Ferguson et al. (2012), 2) entered too large an effect size for Fikkers et al. (2016), and 3) could not be confirmed upon reinspection of Adachi &

Willoughby (2016). The text alludes to "some studies [...] other cases [...] others" but each of these statements appears to refer to a single study. Could you be more precise?

I wonder if it is begging the question to limit your meta-analysis to "the most conservative value available in each study". Do you mean that the smallest estimate from each study was selected? If so, I would expect that to inflict downward bias in the estimates.

#Meta-moderation by research practices

A number of coded best practices need more specific operational definitions:

- How do you determine whether the use of controls is appropriate or inappropriate? That is, what is the operational definition of "theoretically relevant" third variables?
- What is a "standardized" outcome measure?
- What establishes a measure as "clinically validated"?

The authors seem to argue that, because effect sizes are smaller in better studies, the effects are driven by shared method variance. However, I'm not sure that argument can be made given that all these practices are only presented once collapsed together into a best-practices index variable. I'm not sure that the use of ESRB or PEGI ratings is an improvement, since they are coarse (e.g., ESRB ratings have basically only 3 levels: E, T, and M) and influenced by many factors besides violence. To the extent that ESRB/PEGI ratings are less precise measures of violent content, they might decrease the observed effect sizes.

This is a personal thing, but I have difficulty getting excited about citation bias. I do not think citation bias is a good indicator variable for researcher expectancy effects (page 13, line 52). The Cochrane Handbook represents this more as a problem when trying to discover articles, rather than a sign of bias in primary studies (see https://handbook-5-1.cochrane.org/chapter_10/10_2_2_3_citation_bias.htm).

For some analyses it would be helpful to report the conditional average effect size at different levels of these moderators rather than report the correlation between moderator and effect size, which is not easy to convert into those conditional averages.

Please report statistics for nonsignificant results as appropriate, for example at "specific inclusion of theoretically relevant controls was not a predictor of effect sizes" page 11 line 38. Tables would be useful for the exploratory analysis and publication bias sections.

Summary

The estimated .06 is smaller than Prescott's raw crosslag estimate (.11) but not dramatically different from Prescott's reported crosslag w/ covariates (.08). I think the discussion section would be more effective if it interpreted the similarities and explained the differences instead of polemicizing about the asserted meaninglessness of a small effect size.

It is interesting to see that this meta-analysis finds smaller cross-lags at longer intervals, in contrast to the Prescott et al. report. Again, it would be helpful to know why the present study differs from the previous one in this regard.

I appreciate the inclusion of a PRISMA diagram and the availability of the data on OSF.

I always sign my reviews,
Joe Hilgard

Minor points:

I am surprised at the decision to use p-curve only if some proportion of p-values are between .01 and .05. I can understand why one would need some count of p-values between 0 and .05, but I don't understand why one would need some proportion of p-values between .01 and .05.

Page 10, line 35: citation bias is referred to as "publication bias".

Page 11 line 22 and figure 1: It is a "forest plot" not a "Forrest plot".

page 12, line 7 No, p-curve is just fine even if there is not "an unexpectedly high number of statistically significant studies".

page 12, line 17 A citation is needed for "Type II error rates for [procedures such as Egger's test] can be high when effect sizes are very low."

page 12, line 24 "Questionable researcher practices for some studies, as has been documented for the Singapore database." The citation here points to Prescott, Sargent, & Hull (2018); I am not sure I see how they documented QRPs in the Singapore database.

Please post the funnel plots, in supplement if need be.

Page 12, line 50: Please double check the correspondence between Q-stat and p value in "Q = 18.848, p < .151"

Review form: Reviewer 5 (Andrew Przybylski)

Is the manuscript scientifically sound in its present form?

Yes

Are the interpretations and conclusions justified by the results?

Yes

Is the language acceptable?

Yes

Do you have any ethical concerns with this paper?

No

Have you any concerns about statistical analyses in this paper?

No

Recommendation?

Accept with minor revision (please list in comments)

Comments to the Author(s)

In this paper the authors aim to address an important and controversial topic by adopting a relatively novel approach. The question of longer-term violent gaming effects is a valid topic of study and is not well studied in terms of analytic or methodological rigourur. Outsized rhetoric trumps evidence in a number of cases. This is a really interesting study for a few methodological and theoretical reasons. These include the use of a preregistered meta-analytic design, use of SESOI, and focus on longitudinal effects. I am unaware of any studies that have attempted this in the media effects space. Some observations.

1. I think the authors did a good job addressing the comments from the earlier reviewer comments.
2. I think the reviewers could add detail on the preregistration (I checked it, it's fine) but as a methodology and it's potential value for other controversial topics in the behavioural sciences. What went right/wrong that could inform other scholars.

3. I think the SESOI point could be made much more clearly. This might require an explicit framing of superiority/equivalence/inferiority testing for readers who are unfamiliar.
4. I would tone down some of the language/conclusions drawn in the discussion around citation bias and inflation. I think the case is well made for something funny going on but it might not be there for such a strong conclusion as this.
5. The authors might instead use this space to place greater emphasise on the (lack of) evidence for accumulation effects. This is a fine, but important, point that the authors could explain in more detail for readers to consider.

This review is signed,
Andrew K. Przybylski
University of Oxford

Decision letter (RSOS-200373.R0)

Dear Dr Ferguson,

The Subject Editor assigned to your paper ("Do Longitudinal Studies Support Long-Term Relationships Between Aggressive Game Play and Youth Aggressive Behavior? A Meta-Analytic Examination") has now received comments from reviewers. We would like you to revise your paper in accordance with the referee and Associate Editor suggestions which can be found below (not including confidential reports to the Editor). Please note this decision does not guarantee eventual acceptance.

Please submit a copy of your revised paper before 15-May-2020. Please note that the revision deadline will expire at 00.00am on this date. If we do not hear from you within this time then it will be assumed that the paper has been withdrawn. In exceptional circumstances, extensions may be possible if agreed with the Editorial Office in advance. We do not allow multiple rounds of revision so we urge you to make every effort to fully address all of the comments at this stage. If deemed necessary by the Editors, your manuscript will be sent back to one or more of the original reviewers for assessment. If the original reviewers are not available we may invite new reviewers.

When submitting your revised manuscript, you must respond to the comments made by the referees and upload a file "Response to Referees" in "Section 6 - File Upload". Please use this to document how you have responded to each of the comments, and the adjustments you have made. In order to expedite the processing of the revised manuscript, please be as specific as possible in your response.

- Ethics statement

- Data accessibility

<http://datadryad.org/submit?journalID=RSOS&manu=RSOS-200373>

- Competing interests

- Authors' contributions

- Acknowledgements

- Funding statement

Kind regards,
Royal Society Open Science Editorial Office
Royal Society Open Science

on behalf of Dr Christina Demski (Associate Editor)
openscience@royalsociety.org

Associate Editor Comments to Author (Dr Christina Demski):

We have now received additional reviews for this manuscript as well as a second review from a previous reviewer. On balance, the reviewers see merit in this manuscript but would like to see some revisions before it can be considered for publication. In particular, there are some questions about the way the paper is framed and what points are emphasised or not (e.g. citation bias being one aspect that two reviewers have commented on for example).

One reviewer has opted to reject the manuscript and is not satisfied with the revisions that were made. While it is understood that this reviewer was signalled as having a conflict of interest, it is important to get a spread of views from the relevant researchers in the field.

In particular, the reviewer questions how the studies for the meta analysis were chosen. This is an important point that does need to be clarified. The reviewer also questions the cut off value of 0.1, which another reviewer also highlighted as something that needs to be reconsidered in terms of how it is used in the manuscript. Therefore, this point needs further addressing as well. I would also encourage you to engage with the other points the reviewer has made so the editors can be confident the effect sizes that were included in the analysis are accurate.

Reviewer comments to Author:

Reviewer: 1

Comments to the Author(s)

Please see enclosed PDF file

Reviewer: 4

Comments to the Author(s)

The authors present a reproduction and extension of a previous meta-analysis of longitudinal cross-lags from T1 violent video game use to T2 aggressive behavior. I see value in this project, but there are some things the authors could do to make the text more accurate and more agreeable to the reader.

The main claims, as I understand them, are:

- small correlations are not meaningful or are likely to be type I errors
- some effect sizes extracted by Prescott et al. are larger than the original papers would indicate
- studies using better research practices observe smaller effect sizes

#Statistical arguments

I would prefer that you not attempt to argue for a cutoff of .10. I feel that it is begging the question a little bit. I recognize that you cite a source for this, but I doubt that it is broadly appropriate to ignore all effects smaller than $r = .10$. (I am here in disagreement with your previous reviewer 2, who seems to advocate an even higher threshold for some reason.) I would much prefer that the effect size be presented and interpreted on its own terms. I do not think it necessary or helpful to argue that .06 is approximately zero.

I was also very confused at the argument that "Preliminary evidence suggests that the false positive rate for effect sizes below $r = .10$ is very high." I did not find the provided citations helpful on this point. In what domains can you know the true effect size is zero and thus $r = .10$ is mere false positive? Alternatively, if the true effect size is $r = .10$, then that is a true effect. It is again begging the question to assume that a significant effect below .10 (or even just over .10? page 6 line 22) is a false positive rather than a true effect of equal or larger size. It is also not clear

what sort of "noise" you mean in this passage -- sampling error? Meehlian "crud factor"? I think it would be best to cut this line of argument -- the paper is not strengthened by it, and it is not necessary in order to perform your analysis.

Additionally, I must object to the idea of there being such a thing as an "overpowered study" (Page 6, line 6). Render unto statistical significance the things that are statistical significance's, and unto effect size the things that are effect size's.

#Documentation of effect sizes

Some of the contention among the authors, Prescott et al., and their previous reviewer 1 seems to stem from discussions of which effect sizes are correctly scraped or most appropriate. It seems to me one useful way to address and resolve such disagreements would be to 1) note the direct quote or source within each article from which the effect size was scraped (e.g., "Table 2, Model 2, 'T1 VVG'", or "page 13, left column, paragraph 2, $r(183) = .017$ "). Perhaps it would be helpful to see the argument for why an extracted effect size is the most appropriate one. This would be particularly helpful in the case that the effect size extracted differs between Prescott et al. and the current article. Some of this already appears to be listed in the posted response letter, but it would be helpful to make this available to readers and reviewers, especially in the dataset.

I was confused by the passage on page 5, line 45, "However, in some studies...". My best understanding from the passage & provided citations is that you are suggesting that Prescott et al. 1) entered too large an effect size for Ferguson et al. (2012), 2) entered too large an effect size for Fickers et al. (2016), and 3) could not be confirmed upon reinspection of Adachi & Willoughby (2016). The text alludes to "some studies [...] other cases [...] others" but each of these statements appears to refer to a single study. Could you be more precise?

I wonder if it is begging the question to limit your meta-analysis to "the most conservative value available in each study". Do you mean that the smallest estimate from each study was selected? If so, I would expect that to inflict downward bias in the estimates.

#Meta-moderation by research practices

A number of coded best practices need more specific operational definitions:

- How do you determine whether the use of controls is appropriate or inappropriate? That is, what is the operational definition of "theoretically relevant" third variables?
- What is a "standardized" outcome measure?
- What establishes a measure as "clinically validated"?

The authors seem to argue that, because effect sizes are smaller in better studies, the effects are driven by shared method variance. However, I'm not sure that argument can be made given that all these practices are only presented once collapsed together into a best-practices index variable. I'm not sure that the use of ESRB or PEGI ratings is an improvement, since they are coarse (e.g., ESRB ratings have basically only 3 levels: E, T, and M) and influenced by many factors besides violence. To the extent that ESRB/PEGI ratings are less precise measures of violent content, they might decrease the observed effect sizes.

This is a personal thing, but I have difficulty getting excited about citation bias. I do not think citation bias is a good indicator variable for researcher expectancy effects (page 13, line 52). The Cochrane Handbook represents this more as a problem when trying to discover articles, rather than a sign of bias in primary studies (see https://handbook-5-1.cochrane.org/chapter_10/10_2_2_3_citation_bias.htm).

For some analyses it would be helpful to report the conditional average effect size at different levels of these moderators rather than report the correlation between moderator and effect size, which is not easy to convert into those conditional averages.

Please report statistics for nonsignificant results as appropriate, for example at "specific inclusion

of theoretically relevant controls was not a predictor of effect sizes" page 11 line 38. Tables would be useful for the exploratory analysis and publication bias sections.

Summary

The estimated .06 is smaller than Prescott's raw crosslag estimate (.11) but not dramatically different from Prescott's reported crosslag w/ covariates (.08). I think the discussion section would be more effective if it interpreted the similarities and explained the differences instead of polemicizing about the asserted meaninglessness of a small effect size.

It is interesting to see that this meta-analysis finds smaller cross-lags at longer intervals, in contrast to the Prescott et al. report. Again, it would be helpful to know why the present study differs from the previous one in this regard.

I appreciate the inclusion of a PRISMA diagram and the availability of the data on OSF.

I always sign my reviews,
Joe Hilgard

Minor points:

I am surprised at the decision to use p-curve only if some proportion of p-values are between .01 and .05. I can understand why one would need some count of p-values between 0 and .05, but I don't understand why one would need some proportion of p-values between .01 and .05.

Page 10, line 35: citation bias is referred to as "publication bias".

Page 11 line 22 and figure 1: It is a "forest plot" not a "Forrest plot".

page 12, line 7 No, p-curve is just fine even if there is not "an unexpectedly high number of statistically significant studies".

page 12, line 17 A citation is needed for "Type II error rates for [procedures such as Egger's test] can be high when effect sizes are very low."

page 12, line 24 "Questionable researcher practices for some studies, as has been documented for the Singapore database." The citation here points to Prescott, Sargent, & Hull (2018); I am not sure I see how they documented QRPs in the Singapore database.

Please post the funnel plots, in supplement if need be.

Page 12, line 50: Please double check the correspondence between Q-stat and p value in "Q = 18.848, p < .151"

Reviewer: 5

Comments to the Author(s)

In this paper the authors aim to address an important and controversial topic by adopting a relatively novel approach. The question of longer-term violent gaming effects is a valid topic of study and is not well studied in terms of analytic or methodological rigour. Outsized rhetoric trumps evidence in a number of cases. This is a really interesting study for a few methodological and theoretical reasons. These include the use of a preregistered meta-analytic design, use of SESOI, and focus on longitudinal effects. I am unaware of any studies that have attempted this in the media effects space. Some observations.

1. I think the authors did a good job addressing the comments from the earlier reviewer comments.
2. I think the reviewers could add detail on the preregistration (I checked it, it's fine) but as a methodology and it's potential value for other controversial topics in the behavioural sciences. What went right/wrong that could inform other scholars.
3. I think the SESOI point could be made much more clearly. This might require an explicit framing of superiority/equivalence/inferiority testing for readers who are unfamiliar.
4. I would tone down some of the language/conclusions drawn in the discussion around citation bias and inflation. I think the case is well made for something funny going on but it might not be there for such a strong conclusion as this.
5. The authors might instead use this space to place greater emphasise on the (lack of) evidence for accumulation effects. This is a fine, but important, point that the authors could explain in more detail for readers to consider.

This review is signed,
 Andrew K. Przybylski
 University of Oxford

Author's Response to Decision Letter for (RSOS-200373.R0)

See Appendix C.

RSOS-200373.R1 (Revision)

Review form: Reviewer 4 (Joseph Hilgard)

Is the manuscript scientifically sound in its present form?

Yes

Are the interpretations and conclusions justified by the results?

No

Is the language acceptable?

Yes

Do you have any ethical concerns with this paper?

No

Have you any concerns about statistical analyses in this paper?

No

Recommendation?

Major revision is needed (please make suggestions in comments)

Comments to the Author(s)

I thank the authors for accepting the suggestions they did take. They've added some documentation to the raw data to show where these effect sizes came from. They've added a funnel plot. They've made the draft more specific on points that were initially unclear. These are

nice improvements. For me, two issues remain: First, there is still what I feel is a flawed argument for an arbitrary cutoff of $r = .10$ as separating interpretable from uninterpretable effect size. Second, a little more could be done to recognize and credit the strengths of this literature, such as they are, insofar as there does not appear to be publication bias (although I could be misinterpreting these results).

The cutoff at $r = .10$

I still regret that so much of the paper is spent trying to argue that an effect of $r = .06$ is the same thing as an effect of $r = .00$. The authors have tried to soften this point by disclaiming that it will "prompt further discussion on the interpretation of effect sizes," but I feel this is not yet the most productive discussion prompt. I feel that there are many issues just sort of smeared together here: whether the correlation is replicable, whether the correlation represents a causal effect vs. a methodological artifact or confound, and whether the causal effect is large enough to deserve public concern. The authors cite literature describing a number of ways one might go about establishing a smallest effect size of interest on page 18, but do not themselves engage in any of those justifying steps in selecting a cutoff of $r = .10$. Citation xix might help here, but the average correlation between violent video game use and "nonsense outcomes" is $r = .039$, not $r = .10$, so it may speak to a smaller SESOI than the authors chose here.

I also still really dislike the argument that an effect size smaller than $r = .10$ is particularly likely to be a "false positive": "Preliminary evidence suggests that the false positive rate for effect sizes below $r = .10$ is very high, citations x, xix." I don't see how either of these citations supports this claim. Citation x comments that, had those authors not preregistered, they would have been able to cherry-pick an $r = .25$ -- an effect larger than $.10$ that, in context, those authors appear to describe as a possible false positive generated by a high familywise error rate. Citation xix is closer, comparing the effect size to the effect size of various "nonsense outcomes." But this is, again, not an argument about type I error; rather, it is an argument about the smallest effect size of interest. (The same issue applies to the in-progress work of Ferguson & Heene cited in the authors' reply letter.) Again, this muddles together different topics in a way that I do not find helpful, confusing concerns about "crud factor" or other confounding with Type I error.

Maybe my problem is in how the authors use "false", "real", and "true". Let me make it clear: confounds create highly replicable results. Crud may be highly replicable. It is not Type I error to reject the null hypothesis due to crud or to confounds, because in these models, the null hypothesis is false and there is a true effect. These are questions of interpretation and theory and models, not of Type I error. So please do not call these things Type I error.

I think it would be more productive to discuss and interpret the conditional mean for "high quality" studies that we do have, rather than to try to argue that the grand mean effect of $r = .06$ would be $r = .00$ in high quality studies that we have only hypothetically. It may be helpful to consider the confounding of researcher allegiance with preregistration and standardized, validated outcomes.

Adding more detail to Table 2 would help the reader evaluate, and possibly be moved by, the argument that these effects are substantially attributable to flexible quantification or shared method variance. It would be helpful to see the number of studies k , number of participants N , confidence intervals, and heterogeneity statistics.

Publication bias

Table 1 could be formatted a little nicer. For example, in the PET/PEESE row, I don't know what exactly b_0 represents: is it the intercept, or the slope? If it's the intercept, it seems that PET/PEESE is actually estimating the effect size as being larger, right? Similarly, p-curve tests against something two different hypotheses using two or three methods each. Please indicate which one or ones you are reporting. I remember when I looked at the Prescott meta-analysis there were no clear signs of publication bias, consistent with your nonsignificant Egger test and healthy PET/PEESE estimate. If you're testing for publication bias, and you find little evidence of

publication bias, you should say so in the discussion and give it at least one sentence in the abstract, rather than immediately downplaying the relevance of these tests.

When the authors report these publication bias tests, they caution that "When effect sizes are small and homogeneous, and the number of studies relatively small (such as for longitudinal studies of video game violence), power levels are weak even when selection bias is strong (cite xxxvi)." I think they have misunderstood the citation: when citation xxxvi refers to the "range of variances", they mean the sampling errors of studies, not the effect sizes. Homogeneity (as described by low Q , τ , or I^2 values) is likely to improve, not impair, the performance of these tests. Additionally, this citation is about Begg's test, not Egger's test, although what influences one test may well influence the other. It would be nice to have a sentence or two in the results describing the results of PET-PEESE, and it could be useful to add p-curve (available at p-curve.com), p-uniform (available in the puniform package for R), and the three-parameter selection model (available in the weightr package for R).

Minor stuff

Would it be possible to adjust the axes on figures 1 and 2? The x-axes of these plots are quite wide relative to the data, extending all the way to $r = +/-1$ or $Z = +/-2$.

"Our study adds to the observation that effect sizes below $r = .10$ have an increased risk of reflecting methodological noise rather than true effects." To the contrary, I might say that a huge result like $r = 0.9$ suggests some manner of obvious and pernicious confounding. One simply cannot know use a bad estimate to know what would be a good estimate.

I always sign my reviews,
Joe Hilgard

Review form: Reviewer 5 (Andrew Przybylski)

Is the manuscript scientifically sound in its present form?

Yes

Are the interpretations and conclusions justified by the results?

Yes

Is the language acceptable?

Yes

Do you have any ethical concerns with this paper?

No

Have you any concerns about statistical analyses in this paper?

No

Recommendation?

Accept as is

Comments to the Author(s)

The reviewer addressed my outstanding concerns. I thank Dr. Hilgard for raising a couple of important points that I missed.

Decision letter (RSOS-200373.R1)

Dear Dr Ferguson:

Manuscript ID RSOS-200373.R1 entitled "Do Longitudinal Studies Support Long-Term Relationships Between Aggressive Game Play and Youth Aggressive Behavior? A Meta-Analytic Examination" which you submitted to Royal Society Open Science, has been reviewed. The comments of the reviewer(s) are included at the bottom of this letter.

Please submit a copy of your revised paper before 28-Jun-2020. Please note that the revision deadline will expire at 00.00am on this date. If we do not hear from you within this time then it will be assumed that the paper has been withdrawn. In exceptional circumstances, extensions may be possible if agreed with the Editorial Office in advance. We do not allow multiple rounds of revision so we urge you to make every effort to fully address all of the comments at this stage. If deemed necessary by the Editors, your manuscript will be sent back to one or more of the original reviewers for assessment. If the original reviewers are not available we may invite new reviewers.

- Ethics statement

- Data accessibility

- Competing interests

- Authors' contributions

- Acknowledgements

- Funding statement

Kind regards,

Andrew Dunn

on behalf of Dr Christina Demski (Associate Editor)

Associate Editor Comments to Author (Dr Christina Demski):

Associate Editor: 1

Comments to the Author:

Two reviewers from the first round after resubmission have now considered the revised manuscript. While one reviewer is satisfied with the revision the second reviewer still considers some points to be inadequately addressed. As these are quite critical to the overall argument of the paper, we would like for you to provide further responses/revisions on these points.

Reviewer comments to Author:

Reviewer: 4

Comments to the Author(s)

I thank the authors for accepting the suggestions they did take. They've added some documentation to the raw data to show where these effect sizes came from. They've added a

funnel plot. They've made the draft more specific on points that were initially unclear. These are nice improvements. For me, two issues remain: First, there is still what I feel is a flawed argument for an arbitrary cutoff of $r = .10$ as separating interpretable from uninterpretable effect size. Second, a little more could be done to recognize and credit the strengths of this literature, such as they are, insofar as there does not appear to be publication bias (although I could be misinterpreting these results).

The cutoff at $r = .10$

I still regret that so much of the paper is spent trying to argue that an effect of $r = .06$ is the same thing as an effect of $r = .00$. The authors have tried to soften this point by disclaiming that it will "prompt further discussion on the interpretation of effect sizes," but I feel this is not yet the most productive discussion prompt. I feel that there are many issues just sort of smeared together here: whether the correlation is replicable, whether the correlation represents a causal effect vs. a methodological artifact or confound, and whether the causal effect is large enough to deserve public concern. The authors cite literature describing a number of ways one might go about establishing a smallest effect size of interest on page 18, but do not themselves engage in any of those justifying steps in selecting a cutoff of $r = .10$. Citation xix might help here, but the average correlation between violent video game use and "nonsense outcomes" is $r = .039$, not $r = .10$, so it may speak to a smaller SESOI than the authors chose here.

I also still really dislike the argument that an effect size smaller than $r = .10$ is particularly likely to be a "false positive": "Preliminary evidence suggests that the false positive rate for effect sizes below $r = .10$ is very high, citations x, xix." I don't see how either of these citations supports this claim. Citation x comments that, had those authors not preregistered, they would have been able to cherry-pick an $r = .25$ -- an effect larger than $.10$ that, in context, those authors appear to describe as a possible false positive generated by a high familywise error rate. Citation xix is closer, comparing the effect size to the effect size of various "nonsense outcomes." But this is, again, not an argument about type I error; rather, it is an argument about the smallest effect size of interest. (The same issue applies to the in-progress work of Ferguson & Heene cited in the authors' reply letter.) Again, this muddles together different topics in a way that I do not find helpful, confusing concerns about "crud factor" or other confounding with Type I error.

Maybe my problem is in how the authors use "false", "real", and "true". Let me make it clear: confounds create highly replicable results. Crud may be highly replicable. It is not Type I error to reject the null hypothesis due to crud or to confounds, because in these models, the null hypothesis is false and there is a true effect. These are questions of interpretation and theory and models, not of Type I error. So please do not call these things Type I error.

I think it would be more productive to discuss and interpret the conditional mean for "high quality" studies that we do have, rather than to try to argue that the grand mean effect of $r = .06$ would be $r = .00$ in high quality studies that we have only hypothetically. It may be helpful to consider the confounding of researcher allegiance with preregistration and standardized, validated outcomes.

Adding more detail to Table 2 would help the reader evaluate, and possibly be moved by, the argument that these effects are substantially attributable to flexible quantification or shared method variance. It would be helpful to see the number of studies k , number of participants N , confidence intervals, and heterogeneity statistics.

Publication bias

Table 1 could be formatted a little nicer. For example, in the PET/PEESE row, I don't know what exactly b_0 represents: is it the intercept, or the slope? If it's the intercept, it seems that PET/PEESE is actually estimating the effect size as being larger, right? Similarly, p -curve tests against something two different hypotheses using two or three methods each. Please indicate which one or ones you are reporting. I remember when I looked at the Prescott meta-analysis there were no clear signs of publication bias, consistent with your nonsignificant Egger test and healthy

PET/PEESE estimate. If you're testing for publication bias, and you find little evidence of publication bias, you should say so in the discussion and give it at least one sentence in the abstract, rather than immediately downplaying the relevance of these tests.

When the authors report these publication bias tests, they caution that "When effect sizes are small and homogeneous, and the number of studies relatively small (such as for longitudinal studies of video game violence), power levels are weak even when selection bias is strong (cite xxxvi)." I think they have misunderstood the citation: when citation xxxvi refers to the "range of variances", they mean the sampling errors of studies, not the effect sizes. Homogeneity (as described by low Q , τ , or I^2 values) is likely to improve, not impair, the performance of these tests. Additionally, this citation is about Begg's test, not Egger's test, although what influences one test may well influence the other. It would be nice to have a sentence or two in the results describing the results of PET-PEESE, and it could be useful to add p-curve (available at p-curve.com), p-uniform (available in the puniform package for R), and the three-parameter selection model (available in the weightr package for R).

Minor stuff

Would it be possible to adjust the axes on figures 1 and 2? The x-axes of these plots are quite wide relative to the data, extending all the way to $r = +/-1$ or $Z = +/-2$.

"Our study adds to the observation that effect sizes below $r = .10$ have an increased risk of reflecting methodological noise rather than true effects." To the contrary, I might say that a huge result like $r = 0.9$ suggests some manner of obvious and pernicious confounding. One simply cannot know use a bad estimate to know what would be a good estimate.

I always sign my reviews,
Joe Hilgard

Reviewer: 5

Comments to the Author(s)

The reviewer addressed my outstanding concerns. I thank Dr. Hilgard for raising a couple of important points that I missed.

Author's Response to Decision Letter for (RSOS-200373.R1)

See Appendix D.

RSOS-200373.R2 (Revision)

Review form: Reviewer 4 (Joseph Hilgard)

Is the manuscript scientifically sound in its present form?

Yes

Are the interpretations and conclusions justified by the results?

Yes

Is the language acceptable?

Yes

Do you have any ethical concerns with this paper?

No

Have you any concerns about statistical analyses in this paper?

No

Recommendation?

Accept as is

Comments to the Author(s)

No further comments.

Decision letter (RSOS-200373.R2)

Dear Dr Ferguson,

It is a pleasure to accept your manuscript entitled "Do Longitudinal Studies Support Long-Term Relationships Between Aggressive Game Play and Youth Aggressive Behavior? A Meta-Analytic Examination" in its current form for publication in Royal Society Open Science. The comments of the reviewer(s) who reviewed your manuscript are included at the foot of this letter.

Kind regards,

Andrew Dunn

on behalf of Dr Christina Demski (Associate Editor)
openscience@royalsociety.org

Associate Editor Comments to Author (Dr Christina Demski):

Associate Editor: 1

Comments to the Author:

Thank you very much for the revisions. One reviewer who asked for major revisions has now reviewed it again and indicated that the paper can be accepted for publication.

Reviewer comments to Author:

Reviewer: 4

Comments to the Author(s)

No further comments.

Appendix A

Reviewer 1—second review

Conflicts of Interest

The authors begin their rebuttal with a statement that they have no conflicts of interest. They gently suggest that Reviewer 1 may have a conflict as author on the manuscript they criticize. Given the authors' concern, this seems like an opportunity to explore conflicts in somewhat more detail, starting with a definition (Conflicts of Interest in Scientific Research Related to Regulation or Litigation, David Resnik, <<https://www.ncbi.nlm.nih.gov/pmc/articles/PMC2700754/>>: "A researcher has a conflict of interest if and only if he/she has personal, financial, professional, political, or legal interests that have a significant chance of interfering with the performance of his/her or ethical or legal duties."

Financial I assume that when the authors state they have no conflict that this statement means that none have ever benefitted financially from their research on videogames and violence (there is great demand for scientists who are willing to testify on behalf of videogame makers in plaintiff and other lawsuits in the US; industry experts are highly paid). I can state unequivocally that I have never served as an expert on behalf of a plaintiff or videogame maker or served as a consultant in any way to a financial interest related to videogames.

Political I further assume that the authors have no political conflict (that none are against all regulation because of their political beliefs about free speech or government regulation of commercial speech). This is an especially important area of conflict for scientific skeptics. For example, the book *Merchants of Doubt* documents how libertarian physicist Fred Singer has voiced skepticism of science underlying acid rain, global warming, and the notion that cigarettes cause disease, because he rejects government regulation of business.

Politically, I am your typical progressive voter. I believe that it is not unreasonable to pass legislation that keeps violent videogames from being sold or rented to youngsters without parental permission, and that there can be serious debate over the age cut-off for such legislation, but I would be against a ban on violent videogame distribution entirely because that would violate free speech (just as I would be against movie censorship). I admit to being irked that the US Supreme Court struck down California regulations designed to keep violent videogames out of the hands of children, in part because the majority bought the argument that violent videogame effects are too small to be policy relevant, a point of view promoted by the authors. I do not think that my political views unduly influence my opinion about this study.

Professional The authors suggest I have a professional conflict of interest. The notion is that, since they criticize a study I was part of, I am unable to judge their work without bias. Rather than seeing this as a professional conflict, I see it as a difference in the conclusions we draw from a body of scientific work. Ultimately, the authors believe modest effects are likely just "noise" and should be ignored. I believe that small effects are probably real, are likely to be underestimates of the true effect size, and that they can matter. I do not think that is a professional conflict of interest. It is simply a dispute over the science and how it should inform policy.

REVIEW On further careful review, there are major problems with the science that should preclude its publication. The manuscript reads more like an opinion piece.

Problems with Search and PRISMA diagram

The authors haven't explained how they obtained their final list of studies. The authors state "Finally, we undertook a search on PsycINFO and Medline using the terms "(video game*) OR (Computer game*) OR (digital game*)" AND "longitudinal OR prospective" AND "agress* OR viol*" as subject searches, with the exception of longitudinal/prospective which was maintained as all text. This search yielded 50 hits." When I pasted the described search in PubMed I obtained over 130,000 hits. When I pasted it into PsychInfo, I obtained over 140,000 hits. It is impossible to understand how this search strategy contributed to the final number. I wondered if I was doing the search incorrectly so I asked my librarian. Her response wasn't reassuring:

I too retrieved an overwhelming amount of results when I searched that in PubMed. It's possible the author was using filters to reduce the result pool so drastically? They should be indicated if that is the case! This is a mystery to me!

Maybe the authors did not conduct their own search and instead relied mainly on Prescott et al. and simply added studies that they had become aware of; if so, this should be stated explicitly, and the Prescott et al list should be one box on the PRISMA diagram. Finally, the excluded studies boxes do not make it clear what exclusion criteria were being applied at each stage.

RE: "Reviewer 1 requested more information on how effect sizes were extracted for datasets with more than 1 publication. We have now provided more clarity on page 8, second paragraph." The authors state, "Some datasets have produced multiple articles (e.g., the Singapore dataset, also vii and xiii). Given such articles may use different analytical methods, effect sizes can vary considerably between them. Between studies, preference will be given for effect sizes with the maximum number of theoretically relevant controls as well as for preregistered studies should they exist (as was the case for the Singapore dataset). Only one study from each dataset was included in the final analysis."

- Study selection is a process where much bias could be injected; it would be important to be certain that the authors knew nothing about the effect size estimate prior to selecting the study, otherwise they may have been biased in choosing the one with the smallest effect size. In each case where there were multiple publications, the authors need to list all the publications and describe why they chose the one they did. Again, if the authors simply relied on Prescott et al for this process, they should simply state that and not describe a process that was applied only to a couple of new studies identified by the authors.

Persistent Concerns with Preregistration and Best Practices

Reviewer 1 expressed skepticism about "best practices" analyses, referencing Anderson et al. (2010) as an example of another study with some "best practices" analysis. However, an update to that meta-analysis (Hilgard et al., 2017) demonstrated the flawed nature of that specific "best practice" analysis as it had specifically excluded important issues such as standardization, use of proper controls and preregistration. That some have done such analyses poorly in the past (with which we certainly agree) does not diminish the value of a well-conducted best practice analysis. Our best practices analysis focuses on pretty straightforward and widely understood factors associated with high quality research. More importantly, we pre-registered our criteria for the best-practices analysis which limits our abilities to make analytical decisions based on how closely the data supports any pre-existing hypothesis. In short, due to our pre-registration, if the data had have shown larger effects for better designed studies according to our preregistration, we would be bound by that pre-registration to say so. We therefore remain confident in our best practices analysis.

- With respect to the notion that preregistration reduces bias in this case, I am highly skeptical. By my count, this is the fifth time Ferguson has conducted a meta-analysis on this topic. Having done this four times before, he is well aware of the results he is likely to obtain. Preregistration is therefore a low risk procedure. For example, based on his own experience and the results of Prescott, et al, he knows that the combined estimate will be below 0.1. Anderson et al.'s (2010) results for best practices can also be used to inform a best practices designed to decrease the effect size. I'm not saying this is what the authors did, I'm only voicing skepticism that preregistration has much bearing on the scientific validity of this particular study.
 - 1: Ferguson CJ, Kilburn J. The public health risks of media violence: a meta-analytic review. *J Pediatr.* 2009 May;154(5):759-63. doi:10.1016/j.jpeds.2008.11.033. Epub 2009 Feb 23. Review. PubMed PMID: 19230901.
 - 2: Ferguson CJ. The good, the bad and the ugly: a meta-analytic review of positive and negative effects of violent video games. *Psychiatr Q.* 2007 Dec;78(4):309-16. PubMed PMID: 17914672.
 - 3: The effect of academic achievement on aggression and violent behavior: A meta-analysis. Savage, Joanne; Ferguson, Christopher J.; Flores, Lesli; *Aggression and Violent Behavior*, Vol 37, Nov, 2017 pp. 91-101. Publisher: Elsevier Science; [Journal Article]
 - 4: Evidence for publication bias in video game violence effects literature: A meta-analytic review. Ferguson, Christopher J.; *Aggression and Violent Behavior*, Vol 12(4), Jul-Aug, 2007 pp. 470-482. Publisher: Elsevier Science; [Journal Article]

No Support for the 0.1 Partial Correlation Threshold

In the methods, the authors make a poorly referenced argument for a 0.10 cutoff for a partial correlation below which a correlation is not policy-relevant. This is a notion without widespread support in the scientific field, and as will be shown below, is mostly supported by a review article conducted by Ferguson. The attempt to promulgate a policy-relevant threshold is a fatal flaw in this study and in itself is a strong reason to reject its publication. Let's dissect the argument:

“Fourth, the analyses will undertake a more critical examination of effect sizes, employing the cutoff of $r = .10$ suggested by other scholars as an index for interpretable results (Przybylski AK, Weinstein N. Violent video game engagement is not associated with adolescents' aggressive behaviour: evidence from a registered report. Royal Society Open Science. 2019. <https://doi.org/10.1098/rsos.171474>.)

- The notion that the cutoff was suggested by scholars Przybylski and Weinstein is disingenuous. In support of the threshold, Przybylski and Weinstein cite three publications by Ferguson: (1: Ferguson CJ, Kilburn J. 2009 The public health risks of media violence: a meta-analytic review. *J. Pediatr.* 154, 759–763. (doi:10.1016/j.jpeds.2008.11.033) 71. 2: Ferguson CJ. 2013 Violent video games and the Supreme Court: lessons for the scientific community in the wake of *Brown v. Entertainment Merchants Association*. *Am. Psychol.* 68, 57–74. (doi:10.1037/a0030597) and 3: Ferguson CJ. An effect size primer: A guide for clinicians and researchers. *Professional Psychology: Research and Practice.* 2009;40(5):532-538. doi:10.1037/a0015808) as their basis for the cutoff value). The first is a previous meta-analysis that showed that the association was about 0.1. The second is an opinion piece about a Supreme Court decision invalidating a California law prohibiting the sale of violent videogames to minors. Although in both, Ferguson expressed skepticism that a 0.1 correlation is policy-relevant, neither publication proposed such a cut off. The third is a review of effects size metrics, and in this one Ferguson proposes the 0.1 cut-off. Thus, while Przybylski and Weinstein may have been influenced by the publications, but they did not propose the cut off, Ferguson did. Rather than trying to convince us that this cutoff is widely accepted by the scientific community, Ferguson should just own up to it being something he has proposed.
- There are many examples where scientists have called for policy based on correlations of 0.1. One excellent example is childhood lead poisoning. The blood lead IQ partial correlation is about 0.15. Tooth lead may be a better measure of cumulative lead exposure. The partial correlation between tooth lead and IQ is around 0.08. (Needleman HL, Gatsonis CA. Low-level lead exposure and the IQ of children. A meta-analysis of modern studies. *JAMA.* 1990 Feb 2;263(5):673-8. PubMed PMID: 2136923.) Based on longitudinal observational studies, some of which find null results, large scale projects have been undertaken to de-lead public housing and avoid using lead in products that may increase children's lead exposure, like toys.

“Results below this value will be considered non-supportive of a meaningful relationship given their high susceptibility to Type I error (Lobel A, Engels RCME, Stone LL, Burk WJ, Granic I. Video gaming and children's psychosocial wellbeing: A longitudinal study. *Journal of Youth and Adolescence.* 2017;46(4):884–97).”

- The citation used to justify the statement is a study of videogaming and prosocial wellbeing. It says nothing about Type 1 error. The present meta-analysis is mostly about effect size estimates. Type 1 error is about statistical testing. The authors need to explain what they mean by susceptibility to type 1 error and find an appropriate citation. If the citation is the Ferguson Effect Size Primer they need to make it clear that this is their opinion, not some widely accepted fact.

“The use of such a cut-off is valuable to the degree highly powered studies such as meta-analyses may create false positives by identifying methodological noise as ‘statistically significant.’ Preliminary evidence suggests that the false positive rate for effect sizes below $r = .10$ is very high (Przybylski above and Ferguson CJ, Wang JCK. Aggressive video games are not a risk factor for future aggression in youth: A longitudinal study. *Journal of Youth and Adolescence.* 2019;48(8):1439–51).”

Also: Reviewer 1 expressed that prior research has not indicated a preference for “statistically significant” studies in video game research. However, this observation is inaccurate as publication bias has been found in this field several times previous (e.g. Hilgard et al., 2017, Ferguson, 2015). It is likely more difficult to detect in large sample correlational

studies given the small effect sizes involved as we more clearly note in the results section on publication bias on page 11-12.

- I am not aware of a literature that supports the claim that meta-analysis (or large studies) creates false positives. Meta-analysis (and large studies) are able to identify modest effects; then one needs to determine if the effect is relevant from a policy standpoint. That's not a Type 1 error issue. It is about interpretation of the practical importance of the modest effect. The authors need to find a citation independent of their own work that supports the idea that identification of modest effects in meta-analysis (and large studies) represents a type 1 error. Otherwise this statement is not supported.

"It is worth noting that even this cut-off may be generous, and effects higher than .10 may still, in many cases, be spurious. Further research may help elucidate whether .10 or .20 is a more appropriate cut-off though, for now, we'll retain the .10 cut-off. Naturally, any cut-off is somewhat arbitrary ($r = .09$ is little different from .11) but, in this case, reflects a proportional probability of Type I "noise" findings which appear to drop significantly over the .10 threshold (Ferguson and Wang)."

- Please find an independent statistician that agrees with this statement.

"Thus, with effects above .10, scholars can be reasonably confident, in a well-conducted study, that effects are not entirely noise (although noise may still inflate the effect size for the study relative to the effect size for the population. Moreover, this cut-off does not necessarily speak to practical significance: Effects above .10 may still lack practical significance (Ferguson CJ. An effect size primer: A guide for clinicians and researchers. *Professional Psychology: Research and Practice*. 2009;40(5):532-538. doi:10.1037/a0015808)."

- To me that the "noise" is random error and is typically reflected in how wide the confidence interval is. There is actually a strong basis for the idea that when studies have problems with measurement error (all studies that measure both violent videogame exposure and aggression have this problem) meta-analysis tends to UNDERESTIMATE the association. (See *Advances in Methods and Practices in Psychological Science Obtaining Unbiased Results in Meta-Analysis: The Importance of Correcting for Statistical Artifacts* Brenton M. Wiernik, Jeffrey A. Dahlke First Published January 22, 2020 Research Article <https://doi.org/10.1177/2515245919885611> "When constructs are measured with error, the mean effect size in a meta-analysis of relations between measures of these constructs will be biased toward the null (i.e., toward zero for r or d values). The amount of this null-bias in the mean effect size is a function of mean reliability across studies.") Ferguson and colleagues may not believe this, but they need to offer a cogent argument why Wiernik and Dahlke are wrong or withdraw these statements.

Persistent Problems with Extraction of Effect Sizes

Regarding Ferguson, et al,

- I don't view the decision made by Prescott as a mistake. Prescott used the only measure that is consistently reported by Ferguson and colleagues in each of these three studies, the YSR. We appreciate the clarification of methods and have no objection to averaging multiple measures of aggression, which should increase reliability

RE Ferguson et al, "Reviewer 1 questioned whether the Ferguson et al. (2012) and Ferguson et al. (2013) studies are independent."

- This comment mistakenly states that I questioned the independence of the 2012 and 2013 papers from each other. I did not. My concern is that the authors have never made clear the interdependence of the 2011 publication with the 2012 and 2013 publications and hence the reason it was dropped from the current meta-analysis. The comment makes it clear that all three studies were drawn from a single, large 2009 sample. Our review notes that the overlap of the data used in the 2011 paper with those of the 2012 and 2013 studies was never acknowledged in either of the latter publications. Once again, as stated in this meta-analysis, both the 2012 and 2013 papers "cite Ferguson (2011) as an independent source validating the general approach

(Ferguson et al. 2012, p. 142; Ferguson et al. 2013, p. 112) and in correspondence with Prescott et al. the corresponding author represented these findings as independent.” Again, it is the interdependence of the 2011 data with data used in the 2012 and 2013 papers that is at issue along with the failure to acknowledge this in the latter papers; not the fact that each were drawn from a large 2009 sample. It would be helpful to see a flow diagram as a supplement to this meta-analysis that shows clearly how the three samples are independent but drawn from one large 2009 sample, along with selection criteria for each step in order to clear this up once and for all.

Regarding Fikkers et al. (2013).

- Prescott et al. included the Krahe et al. (2012) study after contacting the middle author Robert Busching with a specific request for the video gaming results and receiving the estimates included in the meta-analysis.

Regarding Gentile and Gentile (2008).

- Thanks for clarifying. Two things should be noted about this response. First, in the paragraph on page 134, the authors choose to include a smaller estimate reported for a college student sample ($\beta = .094$), but exclude a larger estimate ($\beta = .229$) reported for 8th and 9th graders included in the same paragraph. Why? This seems like a bias toward the null. Second, and equally important, the result selected from page 135 includes only the direct effects of video game play and ignores its indirect effects. The total effect of a variable includes both its direct and indirect effects. Excluding an effect of video games on an outcome variable that involves its effect via a theoretically meaningful mediator (in this case the impact of video game play on hostile attribution bias) is another example of authors biasing results toward the null.

Regarding Hirtenlehner and Strohmeier (2015) “we obtained the effect size via personal communication with Dr. Hirtenlehner...”

- We have a written response from Dr. Hirtenlehner dated April 24, 2017 in which he provides the estimates we report and confirms our characterization of his study. Do you also have written documentation of your correspondence?

Regarding Krahe et al. (2012),” the effect sizes are indeed obtained from Table 3.”

- Prescott et al received our estimates directly from the author and speculated that the authors averaged values they observed in a table in the article. The authors confirmed this speculation.

Regarding Moller and Krahe, “Reviewer 1 suggested we should include indirect effects as well as direct. However, this is mistaken.”

- I take it that the authors do not appreciate the nature of direct and indirect effects and the manner in which a total effect is recreated from these two components. They seem to imply that we are suggesting that the indirect effect be treated as an independent direct effect and included in the meta-analysis as such. I disagree. They should create and include a total effect comprised of the direct and indirect components. Mediation models are a standard in the field, one we would be reluctant to abandon.

Regarding: “Reviewer 1 questioned our estimate for the Shibuya et al. (2008). We checked it and it is correct. It is reported in table 2 (p. 533) as the relationship between aggression and the interaction between time spent on games with violent video game preferred content. We note that Shibuya only reported the exact data for boys, and our effect size is adjusted for the non-significant outcome for girls as well.

- I have no idea what this means. The authors report an effect of $-.075$ for Shibuya et al. (2008). Above, they say that it is reported in Table 2 on page 533. There is no effect of that size in that table. Moreover, Shibuya et al. did not report any values for non-significant effects, so how is the “effect size adjusted for the non-significant outcome for girls as well”? One is left to assume that the authors took the $-.15$ effect for boys, adopted an effect of $.00$ for girls, and averaged the two. Deriving an estimate in this manner is clearly flawed. As noted in

the previous review, Prescott et al. contacted Shibuya and obtained their estimates directly from her in an e-mail dated July 11, 2014.

RE “Reviewer 1 suggests that if errors were fixed, our effect size would change. This, however, is not true. As noted above, with the few minor fixes required, our overall effect size estimate remained exactly the same ($r = .60$), or would in fact reduce to $r = 0.56$ if Fikkers et al (2013) and Krahe et al (2012) were excluded due to their impure measures of game time.”

- As can be seen from the points made above and below, I do not think that the authors corrected all of the previously identified consequential errors.

RE “Reviewer 1 suggested we should not include data from Teng et al. (2011) because they did not control for game exposure or aggression at time 1.”

- In a longitudinal randomized exposure study, initial states are only controlled in a theoretical sense under an assumption that randomization successfully results in equalization of base rates. Equalization is often not the case, particularly in small studies such as Teng et al. (2010), such that stability in effects creates an illusion of an effect or non-effect. Insofar as the Prescott et al. inclusion criteria required measurement of physical aggression at two points in time, Prescott et al. excluded the Teng et al. study from consideration.

RE: “Reviewer 1 also recommended against including the Wallenius and Punamäki (2008) study, as this study included a Time 2 covariate (parenting communication). Ultimately, we agreed with the rationale of Wallenius and Punamäki (2008) for including this covariate and disagreed with the rationale of Reviewer 1.”

- As noted in the previous review, Wallenius and Punamäki (2008) included a Time 2 covariate (parent-child communication) in their analysis of the effect of Time 1 Game Violence on Time 2 aggression. The latter covariate was the single strongest predictor of Time 2 aggression (other than Time 1 aggression), but has the unfair advantage of being measured coincident with the dependent variable. Despite the author’s confidence in including this estimate, it is prima facie flawed and biases in the direction of the null.

RE: “Reviewer 1 objected to our exclusion of Adachi and Willoughby (2016) due to those authors being unable to provide us an effect size estimate.”

- As with the comments regarding Hirtenlehner and Strohmeier (2015) and Shibuya et al. (2008) above, we communicated directly with the study author in a series of e-mails. We have a written response from Dr. Willoughby dated September 26, 2016 in which she provides the estimates we report.

“We do appreciate this opportunity to double-check our work. At this juncture we are satisfied that our effect size estimates are correct.”

- I am still not satisfied

RE: “Reviewer 1 suggested that our overall effect size might have been lower because of the addition of several Asian samples. Even if this were correct, that would not preclude our inclusion of these samples.

- First, I am not claiming that inclusion of the Asian samples is inappropriate – only that finding weaker effect size estimates are to be expected when one does so. Second, the author’s claim that failure to find a statistically significant moderator effect in a questionable data set is not surprising. Third, Anderson et al. (2010) does not deny the existence of ethnic differences in the effects of violent video games. Indeed, they report significant effects on increased aggressive cognition, decreased prosocial behavior, and decreased empathy in Western relative to Eastern samples, albeit this was observed in correlational studies. For longitudinal studies of aggressive behavior, the larger effect for Western than Eastern studies approached conventional levels of significant ($p < .07$). The latter finding is explicitly noted by Prescott et al. and forms the basis of their test of moderation by race/ethnicity – an hypothesis for which they found statistical support.

RE: “Reviewer 1 suggested that Prescott et al. had examined methodological issues that impact effect size. This statement was incorrect.”

- Prescott et al. did explicitly examine the methodological issue of inclusion or exclusion of covariates on the observed effect sizes and explicitly tested the methodological issues noted by the author (ethnicity, age, study time). It is not clear why the author claims their statement of having examined methodological issues is incorrect.

RE: “Reviewer 1 suggests that including or not including control variables in the effect sizes included in the Prescott et al. meta-analysis had little impact on results.”

- It is not clear what metric the author uses to define “trivial.” If one argues that the observed differences do not constitute “a trivial reduction,” then how can one argue that the substantially larger and statistically significant difference between the observed effects and zero do constitute trivial effects?

Results--Publication Bias

The authors found no evidence of publication bias. Period. Lines 8-29 are interpretive and should be moved to the discussion.

Results--Problems with moderator analysis

RE: “Reviewer 1 questioned the rationale for our analysis given the low effect sizes involved. As we now note toward the bottom of page 5, our hope was to examine how different methodological practices and other issues such as citation bias influence metanalytic results. This has not been previously done in meta-analyses of longitudinal studies of video game violence effects.”

- The authors suggest that the moderator analysis is the big scientific advance in the paper. However, of the 5 reported, two are non-significant, one is characterized by the authors as a “slight trend” and another as “having little impact.”. The final moderator is for “best practices” – a variable critiqued above.
- As Wiernik and Dahlke also point out, measurement reliability also undermines moderator analysis. Rather than biasing toward the null, moderator analyses may have a high Type 1 error rate, especially when the moderator is correlated with measurement reliability: “Effect size heterogeneity and moderator effects. If the studies included in a meta-analysis differ in their measure reliabilities, estimates of the between-studies heterogeneity random-effects variance component (i.e., τ^2 , SD_{res} , SD_p , SD_{δ}) will be artefactually inflated, erroneously suggesting larger potential moderator effects. This most serious if a moderator variable is correlated with measure reliability across studies.” This is a particular problem with the best practices moderator variable, given its reliance on employment of standardized, clinically validated measures and independent ratings of video game content.

Unsupported conclusions in the discussion

Lines	Comment
P13L12-15	This statement needs a citation to support it. If the citation is Ferguson, the statement should be “we believe that...”
P13L17	Statement is incorrect. The cited study found moderation by race/ethnicity with Hispanics showing a significantly lower response to violent videogames.
P13L20	The $r=.10$ “benchmark” is not an accepted “benchmark for interpretability”. As indicated above, this a made up benchmark with little acceptance outside of Ferguson and his circle.
P13L24	Disagree. This meta-analysis demonstrates a modest but potentially important positive relation between violent videogames and aggressive behavior.
P13L29	Preregistration means little here. Ferguson was well aware that this metanalysis would find a combined effect below 0.10, having conducted four similar meta-analyses in the past.

P13L31	The aim of meta-analysis is not primarily to focus on statistical significance. Instead, the aim is to estimate a combined effect based on multiple studies, and to show how much variation there is among studies on effect size estimates. For example the forest plot shows that most studies are either null or modest positive. It is important that NO studies have found a statistically significant negative relation between violent videogame exposure and aggression, which would likely be the case if the association were truly null. Any emphasis on statistical significance is the author's interpretation.
P13L38	Experimental psychology has been slow to adopt confidence intervals, but that's not the case for population researchers, who have tended to focus on the effects estimate. Author shouldn't be able to get away with calling psychologists naïve.
P13L40	This statement is wrong. As pointed out above, measurement error tends to bias meta-analyses such as these toward the null.
P13L45	Methodological noise is random error. This should widen confidence interval, not bias the estimate.
P13,L50	This study did not show that effect sizes are inflated through poorly standardized measures. None of the moderation associations were statistically significant and, as pointed out, measurement error would bias the moderation analysis toward a significant finding.
P13, L54	This study does not address citation bias, so this statement needs a citation.
P14, L8	This study does nothing to show that effect sizes below $r = .10$ have an increased risk of reflecting methodological noise
P14L18	The study found no relation between longer longitudinal periods and effect size. The study did not find a negative association.
P14L33	P values are important in one respect. They can be used to determine whether an investigator may say an association is certain. The author speaks to a certain association in the last comment and this one as well. However in neither case was the moderation association statistically significant. He should not be allowed to state "the use of high quality measures was associated with reduced effect sizes" because his preregistration hypothesis was not confirmed.
P14L38	This study may suggest but does not prove that validated standardized measures was the cause of a moderation effect on ethnicity found by Prescott (it should be noted here that there was a statistically significant moderation effect on ethnicity found in that study). Because the moderation effect by ethnicity was statistically significant in the Prescott paper and the moderation effect was not in this paper, if anything, this would suggest that it is a real ethnicity effect. This hypothesis would best be tested in a single study with a multiethnic sample in which all subjects had the same measures for videogame violence and aggression.

Appendix B

STETSON UNIVERSITY

Department of Psychology

March 2, 2020

Dear Dr. Demski:

I am writing to submit our revised article entitled “Do Longitudinal Studies Support Long-Term Relationships Between Aggressive Game Play and Youth Aggressive Behavior? A Meta-Analytic Examination” for consideration in *Royal Society Open Science*.

Disclosures: This article has not been published previously and is not under consideration elsewhere. There have been no prior presentations of this data. The authors have no conflicts of interest to report. No sponsors were involved in any stage of this manuscript.

We appreciate the consideration that the reviewers have put into their reviews. We do wish to gently note that Reviewer 1 identified themselves in their review as a member of the Prescott et al. (2018) team (though not which specific member). We express the concern that, given our critiques of Prescott et al. (2018) this places that reviewer in a position of having a conflict of interest regarding objectively reviewing our work. We don't intend this as a critique of the editorial process – indeed, having Reviewer 1's review gave us an opportunity to go through our work with their criticisms in mind. This left us with confidence that our paper is robust in light of these criticisms. However, we do express the concern that Reviewer 1 is unlikely to recommend acceptance of our manuscript in any form due to that conflict of interest.

Below we discuss our revisions and replies to the reviewers.

Reviewer 1.

Reviewer 1 suggested we had made a number of mistakes regarding our extraction of effect sizes from individual studies. We appreciated this opportunity to double-check our work. However, in most cases we found that the Reviewer 1 was mistaken in their criticisms (with a few exceptions addressed below and in the revised manuscript). Below we detail our responses and edits to those criticisms.

- Regarding Breuer et al., 2015, Reviewer 1 claimed we had combined two separate “independent” studies into a single datapoint. However, the

reviewer is incorrect. The data all come from a single study with a single sample. Breuer et al., 2015 did conduct some separate analyses stratified by age. However, it is appropriate to maintain the effect size for a single sample as this data collection is a single sample. For confirmation please see Breuer et al. under Participants and Procedures: “After listwise deletion, the final sample (see Data Analysis section) comprised $n = 276$ respondents (i.e., 83% of the subsample)” (Breuer et al., 2015, p. 317)

- The reviewer expressed confusion regarding the Etchells et al./Smith et al. datasets. These studies both use the same dataset, the Avon Heath dataset. Etchells et al., obtained a near-zero effect size (the reviewer focuses on some threshold “statistically significant” effects, but this obfuscates that Etchells acknowledged the effect sizes being trivial and near-zero), whereas with some different analyses and controls, Smith et al. found a slightly negative (but also trivial) effect size. In this case, we had access to the dataset and were able to confirm the slightly negative effect size is most appropriate. We clarify on page 8, first full paragraph, that these two studies use the same dataset.
- Reviewer 1 questioned our effect sizes for the Ferguson et al. (2012) and Ferguson et al. (2013) studies. From our review of our calculations based on this comment it became apparent that the Prescott et al. team made a mistake in extracting effect sizes from these studies. As Reviewer 1 notes, they only used the self-report YSR data from these studies, but these studies also reported outcomes for parent report, bullying and dating violence. Given that Prescott et al. (2013) used similar outcomes for other studies, their use of only the YSR does not present a complete and accurate account of the data presented in these manuscripts. As such, their effect sizes reported for these studies were spuriously inflated. It is important to note that some of these measures of aggression showed smaller effects than the YSR (e.g., the CBCL showed a relationship of $-.03$ compared to the YSR relationship of $+.03$; Ferguson 2012, Table 1, page 143), while others showed a stronger relationship (e.g., the bullying scale showed a relationship of $+.05$ compared to the YSR of $+.02$; Ferguson, 2013, Table 1 page 116). Thus, given the different estimates between these outcomes, we opted to average the effect across the three scales to ensure that all relevant data were represented in the meta-analysis. As this resulted in some larger and some smaller effects being included in our meta-analysis we believe this shows our commitment to ensuring an accurate quantitative account of the literature.
- Reviewer 1 questioned whether the Ferguson et al. (2012) and Ferguson et al. (2013) studies are independent. In fact, they are. The Ferguson et al. (2013) study is a completely independent data collection as is clear from the article as it reports both the T1 and T2 data collections. Ferguson et al. (2012) reports the final timepoint from a study that began in 2009 (both the 2012 T3 outcome study and an earlier 2011 T2 outcome study appropriately and ethnically report being later data

collections from an earlier 2009 correlational study). Contrary to the reviewer's comments, the overlap between the 2011 and 2012 paper was clearly communicated in the 2012 paper "Participants in the current study were drawn from a past project, the Laredo Youth Outcomes Project (Ferguson et al., 2009)." (Ferguson, 2012, p. 142, under participants heading), and in the 2011 paper "Participants in the current study were recruited from a prior study of youth violence (Ferguson et al. 2009)." (Ferguson, 2011, p. 380, under participants heading).

- Regarding Fikkers et al. (2013), Reviewer 1 correctly notes that this article did not clearly distinguish between games and other media (both were included into a composite variable). However, Reviewer 1 failed to note this was also true for an article they included in their own Prescott et al. meta-analysis (Krahe et al., 2012). In both cases we asked the authors for data on games only, but the authors were unable to supply that data. We felt that it was judicious to, nonetheless, include both studies (as Prescott et al., had included the Krahe et al., 2012, study in their own meta-analysis). Both studies received lower best practices scores as they did not have a pure measure of gaming. Excluding both studies would actually reduce the observed effect size slightly to $r = .056$. We have included this new information in second paragraph of the results section.
- Regarding Fikkers et al. (2016), data were obtained directly from the author in personal email correspondence received on June 11, 2019. Any difference in the point estimate is likely attributable to our request for Dr. Fikkers to provide us with a point estimate controlling for gender, time 1 aggression, time 1 aggression norms, and time 1 injunctive norms. This is more control variables than employed in the Prescott et al. (2018) meta-analysis (which only controlled for gender and age), likely explaining any observed difference in the point estimates.
- Regarding Gentile and Gentile (2008), Reviewer 1 expressed being unaware how the effect sizes were obtained. Both are reported directly in the manuscript. Figure 1 (p. 135) was used for the 1st effect size, for the second sample, page 134 second narrative column provides the effect size.
- Regarding Greitemeyer and Sagiogluo (2014), the effect size is reported directly in the manuscript, table 2 (p. 241), path from VVG to trait aggression.
- Regarding Hirtenlehner and Strohmeier (2015) we obtained the effect size via personal communication with Dr. Hirtenlehner on May 6, 2019. It is in fact reported in the manuscript but Dr. Hirtenlehner was kind enough to confirm it was the appropriate effect size. He did not recall having spoken with anyone from the Prescott et al. (2018) team and denied that the effect size they reported in their paper was accurate.
- Reviewer 1 reported minor errors in the effect size and N for Hull et al. (2014). These have now been corrected. They were very small and did not substantively alter any outcomes for the meta-analysis.

- Regarding Krahe et al. (2012), the effect sizes are indeed obtained from Table 3. We are confident these are appropriate for use with our meta-analysis.
- Reviewer 1 noted our reported *N*s were a little off for Lemmens et al., 2011 and Lobel et al., 2017 due to some missing data. We have now fixed these. This did not substantively influence the outcome of the meta-analysis.
- Regarding Moller and Krahe, Reviewer 1 suggested we should include indirect effects as well as direct. However, this is mistaken. Most other studies consider direct effects. Thus, it is inappropriate to include estimates of both direct effects and indirect effects in a meta-analysis as this would amount to comparing very different effects yet treating them as if they are identical. Further, we are concerned that “indirect” effects are often misspecified, and those indirect variables seldom fit the criteria for moderator/mediator variables and, as such, are highly prevalent to Type I error and/or questionable researcher practices (see Fiedler, Schott, & Meiser, 2011 for discussion). We are confident that using only direct effects is the appropriate choice for meta-analysis.
- Reviewer 1 questioned our estimate for the Shibuya et al. (2008). We checked it and it is correct. It is reported in table 2 (p. 533) as the relationship between aggression and the interaction between time spent on games with violent video game preferred content. We note that Shibuya only reported the exact data for boys, and our effect size is adjusted for the non-significant outcome for girls as well.
- Reviewer 1 also noted that missing data would result in lower *N*s for the Staude-Müller (2011) and von Salisch et al. (2011) studies. Reviewer 1 was correct and we fixed these. These minor edits did not substantively change the results of the meta-analysis.
- Reviewer 1 suggests that if errors were fixed, our effect size would change. This, however, is not true. As noted above, with the few minor fixes required, our overall effect size estimate remained exactly the same ($r = .60$), or would in fact reduce to $r = 0.56$ if Fikkers et al (2013) and Krahe et al (2012) were excluded due to their impure measures of game time.
- Reviewer 1 suggested we should not include data from Teng et al. (2011) because they did not control for game exposure or aggression at time 1. This is inaccurate. Teng et al. (2011) involves a longitudinal randomized exposure to game content. Thus, as a randomized condition experiment, initial states are controlled. We therefore felt confident that it was appropriate to retain this study in the meta-analysis.
- Reviewer 1 also recommended against including the Wallenius and Punamäki (2008) study, as this study included a Time 2 covariate (parenting communication). Ultimately, we agreed with the rationale of Wallenius and Punamäki (2008) for including this covariate and disagreed with the rationale of Reviewer 1. We felt confident in maintaining this study.

- Reviewer 1 objected to our exclusion of Adachi and Willoughby (2016) due to those authors being unable to provide us an effect size estimate. In our conversations with both authors, neither confirmed to us that they had spoken to the Prescott group. Adachi and Willoughby were unable to provide us with an effect size estimate. Given other observed errors in the Prescott et al. meta-analysis regarding effect size extraction (including some authors being unable to confirm whether they had spoken to the Prescott et al, group), we were not comfortable accepting a “personal communication” effect size as reported by Prescott without being able to verify that effect size ourselves. Thus, we are confident in excluding this study from our analyses.

We do appreciate this opportunity to double-check our work. At this juncture we are satisfied that our effect size estimates are correct.

Reviewer 1 suggested that our overall effect size might have been lower because of the addition of several Asian samples. Even if this were correct, that would not preclude our inclusion of these samples. However, we tested this concern using moderator analyses as reported in the section “Exploratory Analyses” on page 12. Although there were slight differences between Latinos, Asians and Whites, our moderator analyses found these to be non-significant, suggesting that the Reviewer’s concern is not supported. As we note, particularly for Latino ethnicity, ethnicity was conflated with the use of standardized well-validated instruments such as the Child Behavior Checklist, which are associated with lower effect sizes. Thus, we suspect the evidence better supports that it is better methods rather than ethnicity that explain effect size differences between studies. Indeed, there are no theories in media effects to explain ethnic differences, and some studies have explicitly denied the existence of such ethnic differences (e.g. Anderson et al., 2010).

Reviewer 1 questioned the rationale for our analysis given the low effect sizes involved. As we now note toward the bottom of page 5, our hope was to examine how different methodological practices and other issues such as citation bias influence meta-analytic results. This has not been previously done in meta-analyses of longitudinal studies of video game violence effects.

Reviewer 1 suggested that Prescott et al. had examined methodological issues that impact effect size. This statement was incorrect. Although as Reviewer 1 notes, they did look at issues such as ethnicity, age, and study time, they did not consider methodological issues such as the use of standardized and validated measures, single-responder bias, preregistration, etc. On page 3 in the final lines we clarify that this is what we intended by our use of this terminology.

Reviewer 1 suggests that including or not including control variables in the effect sizes included in the Prescott et al. meta-analysis had little impact on results. Although we agree that effects were weak in all cases, inclusion of control variables in the Prescott et al., meta-analysis reduced random effects outcomes from .106 to .078 or from about

1.2% of variance explained to about 0.6% of variance explained. That's about a 50% reduction in variance explained which is not trivial in our opinion. Our revised estimate of .06 further reduces this effect size by 41% of variance explained; again, we would argue, not a trivial reduction.

Reviewer 1 requested more information on how effect sizes were extracted for datasets with more than 1 publication. We have now provided more clarity on page 8, second paragraph.

Reviewer 1 expressed skepticism about "best practices" analyses, referencing Anderson et al. (2010) as an example of another study with some "best practices" analysis. However, an update to that meta-analysis (Hilgard et al., 2017) demonstrated the flawed nature of that specific "best practice" analysis as it had specifically excluded important issues such as standardization, use of proper controls and preregistration. That some have done such analyses poorly in the past (with which we certainly agree) does not diminish the value of a well-conducted best practice analysis. Our best practices analysis focuses on pretty straightforward and widely understood factors associated with high quality research. More importantly, we pre-registered our criteria for the best-practices analysis which limits our abilities to make analytical decisions based on how closely the data supports any pre-existing hypothesis. In short, due to our pre-registration, if the data had have shown larger effects for better designed studies according to our preregistration, we would be bound by that pre-registration to say so. We therefore remain confident in our best practices analysis.

Reviewer 1 expressed that prior research has not indicated a preference for "statistically significant" studies in video game research. However, this observation is inaccurate as publication bias has been found in this field several times previous (e.g. Hilgard et al., 2017, Ferguson, 2015). It is likely more difficult to detect in large sample correlational studies given the small effect sizes involved as we more clearly note in the results section on publication bias on page 11-12.

As requested by Reviewer 1 we have provided a table of included studies with a link to this table on bottom of page 7/top of page 8.

Reviewer 2

Reviewer 2 noted that the paragraph on page 5 that begins "Therefore, there is value..." was choppy and confusing. It has now been edited for clearer reading.

Reviewer 2 expressed doubt that effect sizes below .20 would likely have practical significance. We agree with the reviewer and expounded our discussion of this in the first paragraph on page 6.

As requested by Reviewer 2 we have added more rationale as to the inclusion of preregistration as a best practice in point #6 on page 9.

Reviewer 2 noted that we had included controlling T1 aggression and gender under both best practices and our “control” outcome. It’s true this does involve a bit of redundancy. It’s inclusion under best practices was conceptualized as kind of a minimum practice. It still applies under “control”, but here we sought to examine a wider range of control variables including these basic controls. We don’t dispute the redundancy, but we do think these variables were important in both analyses.

Reviewer 3

Reviewer 3 asked for more information about the use of the .10 cutoff. We have now expanded our discussion of this issue in the first paragraph on page 6.

Reviewer 3 included several excellent suggestions about reframing our discussion as less a criticism of Prescott et al., and more on the interpretability of tiny effect sizes. The discussion section has been almost entirely rewritten with the Reviewer’s comments in mind. We hope the new discussion which focuses to a greater degree on effect sizes, the low utility of “statistical significance” and other methodological issues will be of greater value.

Reviewer 3 asked about the importance of preregistration of meta-analyses. In response, we have discussed this in the first full paragraph on page 5.

Reviewer 3 asked whether interrater reliability for effect size extraction shouldn’t be 1.00. Unfortunately, many articles include multiple effect sizes often from multiple tests of the same basic hypotheses, creating a situation in which authors must make decisions about which effect sizes are best representative of the hypotheses in question. As a consequence, an interrater reliability of 1.00 would be unexpected for meta-analyses. There are generally some differences which, when identified, can be resolved via dialogue. Were the reliability unusually low this could, of course, mean the two raters have very different assumptions about which hypotheses are being tested. Here, we found adequate interrater reliability. It is of course true that after discussion our final inclusions had an interrater reliability of 1.00.

Reviewer 3 asked whether we might test which specific research practices most impacted effect sizes. We have now added a section for Exploratory Analyses, examining exactly this beginning on page 12 at the bottom.

Reviewer 3 suggested omitting a moderator analysis regarding impact of study inclusion/exclusion from the Prescott et al meta-analysis. This has now been removed.

As requested by Reviewer 3, we have expanded the section on publication bias on page 12, first paragraph.

We believe that these edits address the reviewer concerns and have substantially improved to manuscript. We thank the reviewers for their time and detailed feedback. We look forward to the next round of reviews!

Cordially,

Christopher J. Ferguson, Ph.D.
Department of Psychology
Stetson University
421 N. Woodland Blvd.
DeLand, FL 32729
(386) 822-7288
CJFerguson1111@aol.com

Appendix C

STETSON UNIVERSITY

Department of Psychology

May 12, 2020

Dear Dr. Demski:

I am writing to submit our revised article entitled “Do Longitudinal Studies Support Long-Term Relationships Between Aggressive Game Play and Youth Aggressive Behavior? A Meta-Analytic Examination” for consideration in *Royal Society Open Science*.

Disclosures: This article has not been published previously and is not under consideration elsewhere. There have been no prior presentations of this data. The authors have no conflicts of interest to report. No sponsors were involved in any stage of this manuscript.

We appreciate the consideration that the reviewers have put into their reviews. We do wish to politely note that our concerns about conflicts of interest (CoI) for Reviewer 1 (having identified him/herself as a member of the Prescott et al. team) remain, despite Reviewer 1’s replies to us. His/her review was also unusually long. With that in mind, we are altering the order of our responses, beginning with Reviewer 4, then Reviewer 5, before addressing rebuttals to comments made by Reviewer 1.

Below we discuss our revisions and replies to the reviewers.

Reviewer 4 (Dr. Hilgard).

We appreciate Dr. Hilgard’s transparency in signing his review and will refer to him henceforth by name.

Dr. Hilgard recommended against the .10 cutoff we used for interpretation. As Dr. Hilgard noted, this actually conflicted with the recommendation of Reviewer 2 who recommended a *higher* interpretation threshold and also conflicted with Reviewer 5 (Dr. Przybylski) who asked for more discussion of the .10 threshold (also for the record, Reviewer 3 seemed fine with the .10 threshold whereas Reviewer 1 is against it, though, as we’ll discuss below, arguably for CoI reasons). We’ve taken what is probably a different tact in addressing this than what Dr. Hilgard may have suggested, though we want to assure Dr. Hilgard this was in large part out of respect for his position on this.

Although we may continue to disagree, I hope he will regard our narrative on this as a thoughtful response to his concerns. As such we have taken the following steps.

1) First, we removed mention of the .10 threshold from the abstract which is consistent with Dr. Hilgard's comments.

2) In the remainder of the manuscript, we sought to provide more discussion and rationale for the .10 threshold, from both a theoretical and practical level. Throughout, we have provided more citations for our rationale.

3) As such, we argue our article has additional value...it can function as a prompt for further discussion on the interpretation of effect sizes.

We do not think our discussion is the end of the debate by any means. And we certainly are sure Dr. Hilgard would have many valid counterpoints. However, we hope he will find our attempts here satisfying and be open to dialogue in the public sphere on this important issue.

The new sections are in the introduction, pages 6-7, and in the discussion pages 16-20. We'll reference some specific elements below in response to some of Dr. Hilgard's specific observations.

Dr. Hilgard specifically questioned whether there was evidence that false positives below .10 would be particularly high. We added dialogue on this specific point beginning page 7. In addition, though we could not reference it as it's not yet peer reviewed, one of us (Ferguson) has work in progress with Moritz Heene using the large AddHealth (US) and Understanding Society (UK) datasets that finds very high rates of "statistically significant" results among random variables in these datasets. For AddHealth, depending on analysis type (bivariate v controlled), between 1/2 to 2/3rds of random variable relationships were "statistically significant", with an average effect size of about $r = .043$ (a bit smaller than the .059 effect size we found, but within the confidence interval) though the range of false effects passed over .10. For Understanding Society effect sizes, 8/13 relationships were "statistically significant" despite not having any theoretical basis. The average effect size in this database was actually $r = .11$, so higher than our observed effect. We hope to have this out for review soon. Again, we don't claim this analysis will be the end of the issue, but we do feel it's further evidence there is something of a problem here.

We grant this could be interpreted either through the lens of "false positives" or a more Meehlian "crud factor" (which we acknowledge on page 19, last paragraph), but arguably, neither of these are "good" for how we interpret the meaningfulness of small effects.

We removed the comment about an "overpowered study" from the manuscript. We agree this was poorly phrased.

As suggested by Dr. Hilgard, we have updated the table of effect sizes published on OSF with notes regarding the extraction of effect sizes.

Related to the paragraph beginning “However, in some studies...” on page 4, which Dr. Hilgard found confusing and poorly cited, we have added further examples and sought to clarify the wording.

Under analysis plan, page 10, we sought to clarify what we meant by theoretical controls with appropriate citation. Far from being “downward bias”, understanding the unique role of VVGs in predicting aggression is critical, and employing appropriate controls is critical to this process. In fact, we suspect this process is, if anything, still upwardly biased as few studies actually employed all appropriate theoretical controls.

In response to Dr. Hilgard’s comments on page 11, we provide more detail regarding our coding of best practices.

Dr. Hilgard inquired whether ratings based on ESRB/PEGI were valid indicators of violence. On page 11, #2, we provide evidence for this. ESRB/PEGI ratings have been shown to be valid against blinded rater ratings of violence as well as other approaches used to measure VVG exposure. We provided relevant citations. We also note ESRB actually has a useful 5-point scale, ignoring EC (early childhood), with E (everyone), E10+, T (teen) and M (mature) all getting regular use and AO (adults only) occasionally (though we concede much more rarely than the other 4).

Both Dr. Hilgard and Dr. Przybylski (Reviewer 5) were less than enthused about our citation bias analysis. Because it is in the preregistration we are uncomfortable removing it entirely (we could envision doing so coming afoul of analyses examining matches between preregistrations and final documentation). However, we have greatly scaled back our discussion of it in the results and discussion sections and no longer rely upon it as a main point.

As requested by Dr. Hilgard, we have reported full data for non-significant results where appropriate. We also added Tables 1 and 2, to provide conditional average effect sizes for moderators (Table 2) and all publication bias outcome data (Table 1) as requested.

On page 22, we provide more discussion of our interpretation of the differences in effect size between our study and that of Prescott et al., as suggested by Dr. Hilgard.

As suggested by Dr. Hilgard, we have added discussion regarding our observation that time lags were associated with reduced effect sizes. This is in the “other issues” section beginning page 20.

Dr. Hilgard asked for more details on our rationalization for use of p-curve. We have now provided this at the bottom of page 10.

A funnel plot was added as Figure 1 as suggested by Dr. Hilgard.

Dr. Hilgard mentioned a few minor typos or errors. We don't detail these specifically other than to note these were resolved as Dr. Hilgard suggested.

Reviewer 5 (Dr. Przybylski):

We appreciate Dr. Przybylski transparently identifying himself. We refer to him as Dr. Przybylski henceforth.

Dr. Przybylski suggested adding more discussion related to preregistration, what went right/wrong, etc. We have now done so on page 22 at the bottom.

In response to Dr. Przybylski, we have addressed the SESOI point more clearly, beginning second paragraph page 18. We appreciate Dr. Przybylski bringing our attention to this.

As noted above, we have toned down our coverage of citation bias.

Regarding Dr. Przybylski's comments about the lack of accumulation effects as noted above in response to Dr. Hilgard's comments we did note this in the discussion.

Reviewer 1:

Revisions: We did make a few revisions in response to Reviewer 1. These are described first.

Reviewer 1 inquired why we had only included two of the samples from Gentile and Gentile (2008) and not a third. This was for two reasons. First, the .229 figure Reviewer 1 cited is an *unstandardized* coefficient, not a standardized one, and no standardized coefficient was provided. More critically, it was not a longitudinal sample. However, upon rereading that section we realized the .094 sample was also not longitudinal. The Gentile paper was a bit arcane on this, as it described all the samples and methods together which reduced clarity. However, we realized the .094 effect size was from a correlational sample, not longitudinal and removed it as well. This changed our overall effect size a tiny amount to .059.

As requested by Reviewer 1 we have provided a citation for the sentence that begins "Experimental investigations of the short-term effects..."

Reviewer 1 requested citations by independent statisticians regarding our .10 cutoff. Below, we clarify that the recommendations of Orben and Przybylski are, in fact, independent, even if they happened to cite one of us, as the Orben and Przybylski recommendations are different from those of Ferguson (2009). However, we have also added recommendations by Cohen, Lykken and others to the discussion.

Reviewer 1 stated that she/he was unaware of data that meta-analyses create false positive results. We have now included the data from Kvarven and colleagues that discusses this issue on page 6, final paragraph.

We fixed the citation for the sentence beginning “Results below this value will be considered...” on page 5

Rejoinders: The remainder of our responses to Reviewer 1 take the form of rejoinders to her/his points. No further substantial changes were made.

Reviewer 1 returned to the issue of conflicts of interest. As we stated above, we were not assuaged by the Reviewer’s reply on this matter and, in fact, suggest the reviewer identified a second (political) conflict of interest using his or her own narrative. We mean no disrespect with our comments below. However, by acknowledging his or her authorship on the Prescott et al. meta-analysis, Reviewer 1 has an identifiable professional conflict of interest and we suspect most scholars would agree.

Interestingly, Reviewer 1 discusses political conflicts of interest, which we hadn’t mentioned, noting the case of Fred Singer. Reviewer 1 then acknowledges being “irked” by a US Supreme Court decision on video games and being politically in favor of restricting game sales to minors for “violent” games. Reviewer 1 proceeds to state that he or she does not *think* his or her political views would influence their review, but even that phrasing did not seem certain. As for ourselves, we actually have a divergence of political views within the authorship team, being from different nations, about whether regulation would be an appropriate solution were scientific evidence to link violent games to harm in the real world. Indeed, the first author, who currently resides in the only western country which still employs a Chief Censor, believes that age restrictions of violent content in media is eminently reasonable on a variety of other grounds. They however remain unconvinced of the importance of the typically tiny effects observed in relation to violent contents effects on aggression. Thus, as a team, we do not have a collective identity either opposed to or supportive of regulation to restrict access to violent games.

As such, Reviewer 1, by his or her own acknowledgement, has not only a professional CoI, but also potentially a political CoI. We are not inclined to make too much of the latter under the feeling that all scholars are human. However, we do still argue that the professional CoI is non-trivial.

Our observation on this matter is also supported by the pattern of reviews. We have now obtained 5 independent reviews, 4 of which have been reasonably supportive, even where asking for substantial revisions. Reviewer 1’s reviews have consistently been the most negative and also the longest by far. In fact Reviewer 1’s previous review (at about 4700 words) was actually longer than the article it reviewed (at 4200 words, not including abstract and references). We suggest this as further evidence Reviewer 1 is not entirely dispassionate.

As for ourselves, Reviewer 1 appeared to infer we might have financial ties to the video game industry. We assure Reviewer 1 that none of us do.

Reviewer 1 suggested that, upon trying to rerun our search terms, he received 140,000 hits. As such we reran our search and returned 54. That's a few higher than before, probably because some time has passed since we initially ran the search. It appears likely that Reviewer 1 managed to get 140,000 hits (and we note there are not 140k research studies in any field) by either forgetting to switch "or" commands to "and" or not switching the terms related to aggression or violent games to "subject" searches. Note that these filters were explicitly mentioned in the original manuscript. When we conducted erroneous searches in this manner we get results closer to the 140k figure. Thus, we are confident our search can be replicated and even Reviewer 1 did not appear to suggest we had missed any particular studies.

Reviewer 1 suggested the preregistration was unhelpful because one of us (Ferguson) had conducted several prior meta-analyses and likely knew what effect sizes to expect. With respect to Reviewer 1, this is a non-sequitur. Preregistration is not designed to blind scholars to expected effect sizes (though with any new pool of studies the outcome may not be certain). If that were the case, scholars could never preregister a meta-analysis in any field in which they had conducted prior research. Rather, preregistration is designed to constrain researcher degrees of freedom that can result in false results.

Reviewer 1 also expressed concerns about the .10 threshold. We discussed our changes to the document here in relation to comments by Dr. Hilgard. We believe, at this point, we have thoroughly documented our rationale for this interpretation standard. Reviewer 1 may disagree with us, and we certainly respect that. However, we believe that is an issue for further open debate and dialogue (and one that is extremely important), not a preclusion for publication.

Reviewer 1 continues to misunderstand the original point of longitudinal studies originating from the Ferguson et al (2009) Laredo Longitudinal Project study. To be very clear, Ferguson et al. (2013) bears no relationship whatsoever to this other project and is entirely independent (we had stated this in our last reply as well). Ferguson (2011) and Ferguson et al. (2012) were both follow-ups (at 1 year and 3 years) to the 2009 cross-sectional study, and both clearly state so in their methods sections.

Regarding Krahe et al., 2012 we were unable to confirm Reviewer 1's effect size independently. This is despite contacting the authors (Krahe) ourselves. Note, in this case our reported effect size (.165) is actually *higher* than that reported by Prescott et al (.15) so, if anything, our decision here is inflating, not reducing our effect size.

Regarding Hirtenlehner and Strohmeier (2015), we do indeed have documentation of our personal communication with Dr. Hirtenlehner. We would be happy to send a copy to the editor for validation.

Reviewer 1 suggests we should include effect size estimates that include both direct and indirect effects. This is frankly a serious error. This would introduce serious heterogeneity into any meta-analysis (as most longitudinal studies don't include indirect

effects at all). Further, given that in non-preregistered studies there typically are a host of potential mediating relationships, the potential for questionable researcher practices (e.g. publishing mediation models that “work” and not publishing mediation models that do not) is very high, introducing a clear upward bias into the effect sizes. Further, many mediation models are theoretically ill-defined and supported with mediation/moderation variables serving better as control variables (Savage, 2004).

Reviewer 1 continued to debate our effect size for Shibuya et al., however we are confident in our analysis. We’re not quite sure what data the Prescott et al team requested from Shibuya et al., however, we do note it’s not possible to get their reported effect size based upon the -.15 value for boys and any non-significant value for girls. Although estimating a 0 effect sizes for girls when the non-significant effect size was not reported is not ideal, it is superior to excluding the effect altogether. We did email Dr. Shibuya for clarification but, as of this writing, have not received a reply. Again, we are not confident basing an effect size on a personal communication from Prescott et al., that we were unable to independently verify.

Reviewer 1 noted that they exclude Teng et al. (2011) from their analysis as that study did not meet their inclusion criteria. That’s fine, of course, but the study did meet our inclusion criteria and we are comfortable retaining it. This is the same as Wallenius and Punamäki (2008) for which Reviewer 1 described employing simultaneous controls as “unfair.” We’re not sure what is meant by “unfair” but as current life circumstances such as parental conflict might increase both aggression and a tendency to turn to games to reduce stress, we believe simultaneous controls are entirely appropriate.

Regarding Adachi and Willoughby (2016), those data now appear to be unavailable or lost. Without the ability to independently verify these results ourselves, we are no comfortable using the estimates of Prescott et al. Given that we feel Prescott et al., have made other errors in effect size extraction for other studies, an email exchange between Prescott et al., and Adachi/Willoughby would not be sufficient to assuage our concerns in this matter. Ultimately, it is our professional responsibility to be confident of the effect sizes we use in our analysis, and as such we have to be able to verify an effect size estimate. Without access to the data, we cannot do so.

Reviewer 1 returns to the issue of ethnic differences. In our analyses, ethnic differences were not statistically significant. This is therefore not a valid criticism. Such differences would likely have been problematic anyway because of methodological differences between samples (the high prevalence of standardized well-validated aggression measures among Latino samples in particular).

Reviewer 1 suggests that they did test for methodological issues, mentioning ethnicity, age and study time. None of these, in fact, are methodological issues, they’re demographic variables (with the exception of study time). Here we are discussing issues such as the use of unstandardized measures, or non-independence of video game ratings.

Regarding the issue of substantial reductions in effect size due to the employment of control variables. In the original Prescott et al. meta, their random effects result for bivariate correlations was .106, explaining 1.12% of the variance. For controlled effects, the effect was .078, explaining 0.61% of the variance. That's nearly a 50% reduction in explained variance: clearly not trivial. That's different from considering an overall effect size of .059, which clearly fell below the pre-registered SESOI. Reviewer 1 is, with respect, making an apples and oranges argument.

Reviewer 1 notes that some moderator analyses were non-significant. This is true, but we're not sure what the reviewer's point was here as others were. Reviewer 1 also critiqued the moderator analysis suggesting that heterogeneity statistics might be inflated. However, we tested for moderators directly, not merely relying on heterogeneity statistics and found evidence that some were meaningful. Our analyses merely lend statistical support to that which, frankly, is obvious just looking at the effect size estimates.

Reviewer 1 also mentioned a few sentences he or she did not like. Some of these were redundant with points made above. The only exception was in reference once again to Hispanic samples where the reviewer refers to one of his/her own studies to support ethnic differences (this study was already included in our meta-analysis). As we note on page 21, the utility of this study to be informative is limited by its unstandardized and inconsistent use of aggression measures, and unstandardized video game measures (relying only on three rather random games rather than assessing game play more broadly).

We believe that our revisions address most of the reviewers' concerns. Reviewer 1 is the obvious exception. With no disrespect intended toward Reviewer 1, we do not feel that further back-and-forth with Reviewer 1 is likely to be productive during the review process.

If further revisions are necessary we are, of course, happy to undertake them.

Cordially,

Christopher J. Ferguson, Ph.D.
Department of Psychology
Stetson University
421 N. Woodland Blvd.
DeLand, FL 32729
(386) 822-7288
CJFerguson1111@aol.com

Appendix D

Dear Dr. Demski:

We are writing to resubmit our article entitled "Do Longitudinal Studies Support Long-Term Relationships Between Aggressive Game Play and Youth Aggressive Behavior? A Meta-Analytic Examination." We appreciate the efforts of the final reviewer, Dr. Hilgard and below respond to his comments in a point-and-reply format.

Reviewer Comments

I thank the authors for accepting the suggestions they did take. They've added some documentation to the raw data to show where these effect sizes came from. They've added a funnel plot. They've made the draft more specific on points that were initially unclear. These are nice improvements. For me, two issues remain: First, there is still what I feel is a flawed argument for an arbitrary cutoff of $r = .10$ as separating interpretable from uninterpretable effect size. Second, a little more could be done to recognize and credit the strengths of this literature, such as they are, insofar as there does not appear to be publication bias (although I could be misinterpreting these results).

We thank Dr. Hilgard for raising these issues. Below we address issues around the cut-off of $r = .10$ in more detail.

Regarding Dr. Hilgard's second point, in addition to the existing considerations, in accordance with Dr. Hilgard's suggestions, we now note the absence of publication bias in the manuscript's abstract and in the discussion (p. 15, first paragraph under the discussion).

The cutoff at $r = .10$

I still regret that so much of the paper is spent trying to argue that an effect of $r = .06$ is the same thing as an effect of $r = .00$. The authors have tried to soften this point by disclaiming that it will "prompt further discussion on the interpretation of effect sizes," but I feel this is not yet the most productive discussion prompt.

Although we understand how it may have been perceived, our intent was not to claim that $r = .06$ is the same thing as $r = .00$. By contrast, our point is that, although any non-zero effect can be considered an effect, some effects are small enough to be rendered inconsequential from a practical perspective or may be unreliable enough to give scholars pause in interpreting them. That is particularly the case if methodological weaknesses may drive effect sizes up slightly. To further address this issue, we now incorporate a more specific analysis of the effect size for better quality studies which was $r = .012$, which was statistically indistinguishable from 0.

Nonetheless, we acknowledge that the appropriateness of the $r = .10$ cut-off may of course vary by context, and will, of course, contain a subjective element.

We go to pains in the manuscript to clarify that our suggestion of $r = .10$ as a cut-off is just that: a suggestion, a starting point for discussion. Others will have different views. They have expressed those views and are obviously welcome to continue expressing those views. We seek to contribute to the discussion. In light of our concerns, certainly still part of an ongoing dialogue, we have provided an in-depth consideration of our reasons for (a) believing there is need for a cut-off and (b) selecting .1 as a starting point for consideration. We have now further elaborated on this reasoning (pp.5-6).

Given that other reviewers either agreed with our cut-off or, in one case, even suggested a higher cut-off ($r = .20$) we believe our comments here in the manuscript are an important part of an ongoing dialogue, even if they do not represent a consensus opinion as of yet.

I feel that there are many issues just sort of smeared together here: whether the correlation is replicable, whether the correlation represents a causal effect vs. a methodological artifact or confound, and whether the causal effect is large enough to deserve public concern.

We appreciate the reviewer's concern. We have removed a large section of the manuscript (formerly on p.7-8) in order to better focus our consideration of the utility of a cut-off for practical significance.

The authors cite literature describing a number of ways one might go about establishing a smallest effect size of interest on page 18, but do not themselves engage in any of those justifying steps in selecting a cutoff of $r = .10$. Citation xix might help here, but the average correlation between violent video game use and "nonsense outcomes" is $r = .039$, not $r = .10$, so it may speak to a smaller SESOI than the authors chose here.

Our goal in this paper was not to establish the smallest effect size of interest. Our goal was to present data from a meta-analysis of longitudinal studies about the effects of video game violence on players' aggression. However, we also wanted to consider these outcomes with reference to a defensible cut-off for considering an effect too small for practical relevance, and to promote further consideration of this issue in the literature.

Thus, we did not engage in a variety of steps for determining the smallest effect size of interest. We selected and justified a cut-off suitable to begin the conversation. Importantly we pre-registered this as the smallest effect we were comfortable interpreting ahead of time.

The amount of space spent on the justification for this choice has necessarily become larger – and, therefore, the issue more prominent in the manuscript – at the request of reviewers.

It is worth noting that in Ferguson and Wang (2019) although the mean effect size of nonsense variables was .039, the range included effect sizes that exceeded .10 (up to .144). In other works currently under review (Ferguson & Heene, 2020), the average effect size was close to (in one dataset) .10 or exceeded it (in a second dataset). Therefore we are confident that .10 is a reasonable benchmark.

I also still really dislike the argument that an effect size smaller than $r = .10$ is particularly likely to be a "false positive": "Preliminary evidence suggests that the false positive rate for effect sizes below $r = .10$ is very high, citations x, xix." I don't see how either of these citations supports this claim. Citation x comments that, had those authors not preregistered, they would have been able to cherry-pick an $r = .25$ -- an effect larger than .10 that, in context, those authors appear to describe as a possible false positive generated by a high familywise error rate. Citation xix is closer, comparing the effect size to the effect size of various "nonsense outcomes." But this is, again, not an argument about type I error; rather, it is an argument about the smallest effect size of interest. (The same issue applies to the in-progress work of Ferguson & Heene cited in the authors' reply letter.) Again, this muddles together different topics in a way that I do not find helpful, confusing concerns about "crud factor" or other confounding with Type I error.

With changes to wording (around Type I error) and the removal of some sections of the manuscript (detailed above), we believe this concern has been addressed.

We have, however, also included an additional citation elsewhere to support a related claim in the literature about the relationship between small effect sizes and increased risks of "false positive": Ioannidis, J. P. (2005). Why most published research findings are false. PLoS med, 2(8), e124.

Maybe my problem is in how the authors use "false", "real", and "true". Let me make it clear: confounds create highly replicable results. Crud may be highly replicable. It is not Type I error to reject the null hypothesis due to crud or to confounds, because in these models, the null hypothesis is false and there is a true effect. These are questions of interpretation and theory and models, not of Type I error. So please do not call these things Type I error.

We appreciate the issue the reviewer raises here. We used terms like "true" and "real" as shorthand to refer to the effects of interest (as opposed to noise). Often, we used inverted commas to signify that this terminology is somewhat artificial. Similarly, we used Type I error as shorthand for "incorrectly rejecting the null in favour of the effect of interest" rather than "incorrectly rejecting the null in favour of an effect genuinely present in the data".

We have modified our terminology throughout to be clearer on this point. In this sense we have generally replaced terminology such as "false results" or "false positive" with

“misleading results.”

I think it would be more productive to discuss and interpret the conditional mean for "high quality" studies that we do have, rather than to try to argue that the grand mean effect of $r = .06$ would be $r = .00$ in high quality studies that we have only hypothetically. It may be helpful to consider the confounding of researcher allegiance with preregistration and standardized, validated outcomes.

We thank Dr. Hilgard for this suggestion. As recommended, we now include analyses in the exploratory analyses section, final paragraph, examining studies that were above the median (scoring 3 or above) for best practices. We also noted these results briefly in the abstract. Also in the discussion under “other issues” we added a brief discussion of the best practices results as well as noting the overlap between researcher allegiances and some best practices,

Adding more detail to Table 2 would help the reader evaluate, and possibly be moved by, the argument that these effects are substantially attributable to flexible quantification or shared method variance. It would be helpful to see the number of studies k , number of participants N , confidence intervals, and heterogeneity statistics.

As recommended, we have adjusted Table 2 with the additional information.

Publication bias

Table 1 could be formatted a little nicer. For example, in the PET/PEESE row, I don't know what exactly b_0 represents: is it the intercept, or the slope? If it's the intercept, it seems that PET/PEESE is actually estimating the effect size as being larger, right? Similarly, p-curve tests against something two different hypotheses using two or three methods each. Please indicate which one or ones you are reporting.

As recommended, Table 1 was adjusted to indicate that the value for PET/PEESE involved the intercept and p-curve reported statistics were testing for lack of evidentiary value due to p-hacking.

I remember when I looked at the Prescott meta-analysis there were no clear signs of publication bias, consistent with your nonsignificant Egger test and healthy PET/PEESE estimate. If you're testing for publication bias, and you find little evidence of publication bias, you should say so in the discussion and give it at least one sentence in the abstract, rather than immediately downplaying the relevance of these tests.

We have removed the section “downplaying” the results, and included explicit acknowledgement of the findings in the abstract and discussion.

When the authors report these publication bias tests, they caution that "When effect sizes are small and homogeneous, and the number of studies relatively small (such as for longitudinal studies of video game violence), power levels are weak even when selection bias is strong (cite xxxvi)." I think they have misunderstood the citation: when citation xxxvi refers to the "range of variances", they mean the sampling errors of studies, not the effect sizes. Homogeneity (as described by low Q , τ , or I^2 values) is likely to improve, not impair, the performance of these tests. Additionally, this citation is about Begg's test, not Egger's test, although what influences one test may well influence the other.

We have removed this problematic section.

It would be nice to have a sentence or two in the results describing the results of PET-PEESE, and it could be useful to add p-curve (available at p-curve.com), p-uniform (available in the puniform package for R), and the three-parameter selection model (available in the weightr package for R).

At the end of the paragraph under Publication Bias we include a brief description of the PET-PEESE results.

Minor stuff

Would it be possible to adjust the axes on figures 1 and 2? The x-axes of these plots are quite wide

relative to the data, extending all the way to $r = +/-1$ or $Z = +/-2$.

As requested the scale of both figures has now been adjusted.

"Our study adds to the observation that effect sizes below $r = .10$ have an increased risk of reflecting methodological noise rather than true effects." To the contrary, I might say that a huge result like $r = 0.9$ suggests some manner of obvious and pernicious confounding. One simply cannot know use a bad estimate to know what would be a good estimate.

This sentence has been removed.

We hope that these revisions address the reviewer's comments.

Cordially,

Aaron Drummond
James Sauer
Christopher J. Ferguson